# Biomass burning CO, PM and fuel consumption per unit burned area estimates derived across Africa using geostationary SEVIRI Fire Radiative Power and Sentinel-5P CO data

Hannah M. Nguyen[1,2], Jiangping He[1,3], Martin J. Wooster[1,2,3]

[1]Department of Geography, King's College London, London, WC2R 2ND, UK
[2]Leverhulme Centre for Wildfires, Environment and Society, UK
[3]National Centre for Earth Observation (NCEO), UK

*Correspondence to*: Hannah M. Nguyen (hannah.nguyen@kcl.ac.uk)

**Abstract**: We present the first top-down CO fire emissions inventory for Africa based on the direct relation between geostationary satellite-based Fire Radiative Power (FRP) observations and polar orbiting satellite observations of Total Column Carbon Monoxide (TCCO). This work significantly extends the previous Fire Radiative Energy Emissions (FREM) approach that derived Total Particulate Matter (TPM) emission coefficients from FRP and Aerosol Optical Depth (AOD) observations. The use of satellite-based CO observations to derive CO emission coefficients, $EC_{CO}^b$, addresses key uncertainties in the use of AOD observations to estimate fire-generated CO emissions including; the requirement for a smoke mass extinction coefficient in the AOD to TPM conversion; and the large variation in TPM emission factors - which are used to convert TPM emissions to CO emissions. We use the FREM-derived CO emission coefficients to produce a Pan-African CO fire emission inventory spanning 2004 to 2019. Regional CO emissions are in close agreement with the most recent version of GFED(v4.1s), despite the two inventories using completely different satellite datasets and methodologies. Dry Matter Consumed (DMC) and DMC per unit burned area are generated from our CO emission inventory – the latter using the 20 m resolution Sentinel-2 FireCCISFD burnt area (BA) product for 2019. We carry out an evaluation of our FREM-based CO emissions by using them as input in the WRF-CMAQ chemical transport model and comparing simulated TCCO fields to independent Sentinel-5P TROPOMI TCCO observations. The results of this evaluation show FREM CO emissions to generally be in good agreement with these independent measures - particularly in the case of individual fire-generated CO plumes where modelled in-plume CO was within 5% of satellite observations with a coefficient of determination of 0.80. Modelled and observed total CO, aggregated over the full model domain, are within 4% of each other, though localised regions show an overestimation of modelled CO by up to 50%. When compared to other evaluations of current state-of-the-art fire emissions inventories, the FREM CO emission inventory derived in this work shows some of the best agreement with independent observations. Updates to previously published FREM TPM emissions coefficients based on this methodology are also provided, along with a similar evaluation as conducted for CO. The methodology described in this work is forming the basis of a forthcoming near-real-time fire emissions product from Meteosat to be issued by the EUMETSAT LSA SAF (https://landsaf.ipma.pt/en/).

# 1    Introduction

The open burning of biomass in landscape fires is amongst the largest contributor of gaseous and particulate emissions to the atmosphere. In many regions such fires show significant interannual variability, and together global biomass burning generates a significant fraction of many atmospheric species, including for example the pollutants: total particulate matter (TPM) and carbon monoxide (CO) (Andreae and Merlet, 2001; Bowman et al., 2009; Forster et al., 2007). Landscape fire emission inventories are thus essential to many studies in Earth system science (Keywood et al., 2013; Langmann et al., 2009), and also to "real-time" decision-making applications such as air quality forecasting. Fire emissions inventories are often constructed using a 'bottom-up' approach in which estimates of dry matter consumed (DMC) are estimated from satellite-derived metrics of burned area (BA), or occasionally active fire (AF) counts, combined with information on pre-fire fuel load and combustion completeness (Seiler and Crutzen, 1980). The resulting DMC estimates are then multiplied by biome-specific emission factors (EFs) that relate each kilogram of burned biomass to the amount of a trace gas or aerosol species released. EFs are typically derived from small scale laboratory or ground-based field measurements (Akagi et al., 2011; Andreae, 2019; Andreae and Merlet, 2001), along with airborne sampling of fire plumes (Abel et al., 2003; Lavorel et al., 2007; Quennehen et al., 2012). The Global Fire Emissions Database (GFED) is the most widely-used 'bottom-up' fire emissions inventory (van der Werf et al., 2006, 2010, 2017), but the reliance on burned area and pre-fire fuel load information means it cannot provide near real-time information. The Global Fire Assimilation System (GFAS) (Kaiser et al., 2012) uses near-real time satellite observations of AFs to drive its DMC estimates, performing the conversion using a previously-derived calibration relationship between a given biome's mean Fire Radiative Energy (FRE) and DMC totals from GFED over a specific period. The primary advantage of GFAS is the near real-time aspect, capable of delivering information suitable for driving atmospheric models in forecast mode. The main disadvantage is the fact that the relatively uncertain fuel load and combustion completeness assumptions, which introduce some of the most significant uncertainty to bottom-up fire emissions calculations, are also incorporated into GFAS via the calibration with GFED (Kaiser et al., 2012; Reid et al., 2009; Zhang et al., 2008). Other global fire emissions inventories such as FLAMBE (Reid et al., 2009) and FINN (Wiedinmyer et al., 2011) contain aspects of the same methodologies and thus suffer similar uncertainty sources.

Recently so-called 'top-down' fire emissions methodologies have evolved, partly in an attempt to remove the limitations induced by calibrating satellite-derived FRE observations against DMC totals produced by e.g. GFED. These 'top-down' methodologies include the Fire Energetics and Emissions Research (FEER) approach of Ichoku and Ellison (2014) and Fire Radiative Energy Emissions (FREM; Mota and Wooster, 2018; Nguyen and Wooster, 2020). The FEER and FREM approaches derive landscape fire emissions estimates directly from satellite-derived FRE, thereby removing the step requiring calculation of DMC and thus the uncertainties inherent in that calculation. In each method, a scalar (a so-called "smoke emission coefficient"; $EC_x^b$ in g.MJ$^{-1}$) is generated for each fire-affected biome, $b$, to capture the relationship between the rate of FRE emission (i.e. the fire radiative power [FRP] of the causal fire) and the associated emission rate of a particular trace gas or aerosol species, $x$. Importantly, because $EC_x^b$ values derived from laboratory fire measurements (e.g. as per Freeborn et al.,

2008) may not be fully representative of all the effects relevant to satellite data of real landscape fires (Freeborn et al., 2008; Mota and Wooster, 2018), the FEER and FREM approaches instead used $EC_x^b$ values derived from satellite datasets. Specifically, individual fire matchups where the fires' radiative energy emissions and its smoke plume observed from satellite are used to generate the $EC_x^b$ values. Thus far, both FEER and the FREM approaches have focused on smoke plume

observations of aerosol optical depth (AOD), which are used to create in-plume values of total particulate matter (TPM) via application of a smoke aerosol mass extinction coefficient, $\beta_e$ (in $m^2.g^{-1}$) - as described by Ichoku and Ellison (2014) and Nguyen and Wooster (2020). One key difference between the FEER and FREM approaches is that in FEER per pixel $EC_x$ coefficients are generated from fire match ups within a 1x1 grid, whereas in FREM biome-specific emission coefficients are generated $[EC_x^b]$ where $b$ denotes the biome. Once representative $EC_{TPM}^b$ values are obtained using this matchup dataset, they

can be applied to the FRP data of any fire to derive its rate of TPM emission – including from near real-time satellite data feeds. Although uncertainties in the DMC conversion step are removed in these top-down approaches, other uncertainties are introduced - primarily from uncertainties in the satellite-derived datasets, and in the case of $EC_{TPM}^b$, the use of the mass extinction coefficient, $\beta_e$, used in the conversion of AOD to TPM. The FEER approach of Ichoku and Ellison (2014) uses polar orbiting MODIS data to provide the FRP records driving its TPM emissions estimates, whilst FREM uses the higher-

temporal-resolution FRP data available from geostationary satellites. The latter provides the highest temporal frequency TPM emissions estimates currently available for Africa (Nguyen and Wooster, 2020), and this type of high frequency emission information has been shown useful for maximising the accuracy of smoke transport modelling (Baldassarre et al., 2015; Garcia-menendez et al., 2014). The use of geostationary FRP data with FREM also allows a simple temporal integration to be used to calculate FRE (see Nguyen and Wooster (2020)), obviating the need for assumptions about the plume height, wind speed and

wind direction as is used by FEER when deriving $EC_{TPM}$ from individual MODIS FRP observations (Ichoku and Ellison, 2014). A drawback of using geostationary AF data is that, at present, operational geostationary satellites have a lower spatial resolution than do polar-orbiting sensors, resulting in the under-detection of 'small' or low-FRP fires. This 'missing' contribution to the FRE can be accounted for using a so-called 'small fire correction' (discussed in more detail in Section 2.1).

The purpose of the current work is to adapt the FREM approach further to derive trace gas emissions estimates directly from FRP observations, without first estimating a $EC_{TPM}^b$ emissions coefficients as a precursor. The Mota and Wooster (2018) and Nguyen and Wooster (2020) iterations of FREM both estimated emissions of a target trace gas, $y$ (e.g. $CO_2$, $CH_4$ or $CO$), from the biome-specific emissions coefficient, $EC_y^b$, which was itself derived via Equation 1 using the emission coefficient of a reference species, $x$ (thus far always TPM):

$$EC_y^b \, [g.MJ^{-1}] = \frac{EF_y^b \, [g.kg^{-1}]}{EF_x^b \, [g.kg^{-1}]} \cdot EC_x^b [g.MJ^{-1}] \qquad [1]$$

Here, $EC_y^b$ is the emissions coefficient for the target species $y$ (e.g. CO) in biome, $b$, $EF_y^b$ is the target species $y$ emission factor in that biome, $EF_x^b$ is the emission factor for the reference species, $x$, in that biome, and $EC_x^b$ is the FREM-based emissions coefficient for the references species, $x$, in that biome.

Using Equation 1 to translate between emissions coefficients does introduce some uncertainty, mainly due to the emissions factors of the reference species used thus far (TPM) typically being far from constant even in a single biome, for example, $EF_{TPM}$ is relatively poorly constrained in tropical forest and cultivated land (Akagi et al., 2011; Andreae, 2019). Here we aim to directly generate CO emissions coefficients [$EC_{CO}^b$] by replacing the matchup fire plume AOD information currently used by FREM (Nguyen and Wooster, 2020) with that of total column CO (TCCO in mol.m$^{-2}$) from Sentinel-5P TROPOMI

observations (Landgraf et al., 2016). CO concentrations in landscape fire plumes are higher than in the ambient atmosphere (e.g. Wooster et al., 2011), thus providing potential for distinct contrasts between a smoke plume and its background in the Sentinel-5P TCCO record. A further advantage of generating $EC_{CO}^b$ values directly, rather than via Equation 1, is that it removes the requirement for the smoke aerosol mass extinction coefficient ($\beta_e$), used to generate TPM estimates from AOD measure. The $\beta_e$ coefficient itself is somewhat dependent on fuel type burned, smoke aging and atmospheric relative humidity (Chin et

al., 2002; Formenti et al., 2003; Reid et al., 2005) therefore direct use of satellite TCCO retrievals to derive $EC_{CO}^b$ removes this uncertainty source from FREM-derived estimates of trace gas emissions coming from the satellite FRP retrievals.

## 2     Methodology

### 2.1     FRP and CO Datasets

Africa is the most fire affected continent on the planet (van der Werf et al., 2017), and to derive $EC_{CO}^b$ values we focused on

obtaining matchup fires across Africa's fire-affected biomes as observed by the geostationary Meteosat SEVIRI instrument and by the polar-orbiting Sentinel-5P (S5P) TROPOMI sensor. The Meteosat FRP-PIXEL (FRP in MW) product is generated every 15-minutes from SEVIRI observations and is issued in near real-time by the EUMETSAT LSA SAF (https://landsaf.ipma.pt/en/data/catalogue/). The offline (OFFL) S5P total column carbon monoxide (TCCO in mol.m$^{-2}$) product used in this work can be downloaded from Sentinel-5P Pre-Operations Data Hub (https://scihub.copernicus.eu/) and

has a spatial resolution of $7 \times 7$ km until August 2019 after which the along-track resolution was increased to 5.5 km.

Under cloud-free conditions, which predominate during African fire seasons, the FRP-PIXEL product provides almost continuous landscape fire observations. The coarser pixel size of geostationary observations mean they have a higher minimum FRP detection limit than do polar-orbiting FRP datasets such as those from MODIS and VIIRS (Roberts et al., 2005, 2015)

and will therefore detect fewer 'small' AFs and hence measure less instantaneous FRP than these polar orbiting sensors. The AFs detectable in the geostationary FRP products still remain significantly smaller in terms of pixel area coverage (e.g. down to perhaps 0.01% of the pixel) than the minimum burned area detectable in the MODIS burned area products, however - these are the most common data source of bottom-up fire emission estimation approaches (Van Leeuwen et al., 2014; Reid et al., 2009; Vermote et al., 2009). A recent comparison between AFs detected by the 30 m spatial resolution Landsat-8 Operational

Land Imager (OLI) and Meteosat FRP-PIXEL data showed the geostationary product to have an 8% error of commission, 'false alarm rate' (Hall et al., 2019), very similar to that of the widely used MODIS AF products (Giglio et al., 2016). Prior comparisons between the SEVIRI FRP-PIXEL product and the 1 km MODIS MOD14 product have identified the FRP-PIXEL product's AF error of omission rate and rate of FRP underestimation compared to MODIS (Roberts et al., 2015; Wooster et al., 2015). Region-specific mean 'small fire' scaling factors were derived from comparisons of coincident and co-located

aggregated FRP from the MODIS and SEVIRI active fire products. These were determined to be 1.67 and 1.46 for Northern and Southern Hemisphere Africa (NHAF and SHAF) respectively, and following Mota and Wooster (2018) these factors were applied to account for the average amount of FRP coming from fires burning below the SEVIRI sensor's minimum FRP detection limit. We also apply the cloud cover correction used in the LSA SAF Meteosat FRP-GRID product (Wooster et al., 2015),  though the effect of this adjustment is limited due to sparse cloud cover during the African fire season. To aid

identification of suitable fire matchups we also use visual band data from the Visible/Infra-red Imager and Radiometer Suite (VIIRS) onboard the Suomi-NPP (National Polar-orbiting Partnership) satellite. Suomi-NPP overpasses within 3.5 minutes of the Sentinel-5P overpass and provides 375 m and 750 m spatial resolution imagery that greatly benefits plume identification in the TROPOMI CO data. VIIRS imagery data were obtained from https://ladsweb.modaps.eosdis.nasa.gov/.

## 2.2    Top-down FREM-CO Methodology

As introduced in Section 1, the FREM methodology derives a biome-dependent 'smoke emission coefficient' for a reference species $x$ $[EC_x^b]$ from the relationship between the thermal energy a fire radiates (i.e. the FRE in MJ) and the mass of the reference compound $x$ emitted (in kg or g) over the same time period. Focusing on CO, we derived $EC_{CO}^b$ values from a set of matchup fires for which good observations of both FRE from the fire and TCCO from its plume exist. The derived $EC_{CO}^b$ has

units of g.MJ$^{-1}$ or g.s$^{-1}$.MW$^{-1}$ and can be used to generate CO emission rates (or totals) for a fire when multiplied by the FRP (or FRE) estimate for that fire. Sentinel-5P TROPOMI (S5P) TCCO retrievals are available from May 2018 until the present day, and we gathered our matchup data from joint S5P TCCO, Meteosat FRP-PIXEL, and VIIRS RGB imagery covering July to December 2018 and the full year of 2020. We studied both Northern and Southern Hemisphere Africa (NHAF and SHAF), which have asynchronous fire seasons. We derived $EC_{CO}^b$ values for the six 'fire biomes' of Africa mapped by Nguyen and

Wooster (2020), with this mapping based on re-classification of a 2019 landcover dataset generated from 300 m spatial resolution MERIS and PROBA-V observations as part of the European Space Agency (ESA) Climate Change Initiative (CCI) (https://cds.climate.copernicus.eu/cdsapp#!/dataset/satellite-land-cover). To provide further biome discrimination for *woodland savanna/open forest*, we made use of percentage tree cover information (above 5 m height), taken from a 2015 map of Vegetation Continuous Fields (VCF) generated from 30 m Landsat data (https://landsat.gsfc.nasa.gov/). $EC_{CO}^b$ values were

generated for the resulting six "fire biomes" - *closed canopy forest*, *low-woodland savanna/open forest, high-woodland savanna/open forest*, *grassland*, *shrubland* and *managed land*.

Plume selection was carried out based on S5P TCCO observations (mol.m$^{-2}$) and near co-incident VIIRS imagery (**Figure 1**). Each match-up fire was selected and filtered by manually defining a polygon that encapsulated the smoke plume and responsible AF pixels. There were several criteria that match-ups had to comply with in order to be included for $EC_{CO}^b$ derivation:

- a relatively short period since the start of the fire (mean of 3.4 hours since first AF detection) to minimise plume dispersion and thus maintain an optimal background TCCO to in-plume TCCO ratio for calculating the fire's TCCO anomaly, this temporal limit also reduced the opportunity for oxidation of CO to affect these data;
- no other fires being present in the area during the day/days preceding the match-up fire being identified;
- some wind-driven dispersion of the plume to minimise the chance of thick smoke generated by the fire covering the location of some of the AF pixels and causing the area containing FRP to be incorrectly masked as cloud;
- cloud free observations of the fire and plume throughout the lifetime of the fire, determined from the FRP-PIXEL Quality Product (3 km at nadir) detailed in Wooster et al. (2015) and the visible band VIIRS imagery (750 m at nadir)

The S5P TCCO product also includes a quality value (qa_value) indicating the potential presence of cloud in the total column of air from which a CO retrieval is made. When a threshold of qa_values > 0.5 was applied to the S5P data this resulted in an average difference of only 2.4% in the derived $EC_{CO}^b$ values. Furthermore, the validation of the qa_values assigned in the S5P TCCO product was carried out in clear-sky only conditions (Borsdorff et al., 2014). The S5P qa_values were therefore not implemented in the $EC_{CO}^b$ derivation detailed below, a full discussion of this is detailed in Appendix A. A buffer of pixels surrounding each plume was included in each manually defined polygon such that the buffer easily encapsulated the high TCCO value pixels of the plume, plus a series of pixels outside of the plume from which the 'background' CO amount relevant to each plume was calculated. Only match-up fires for which a single 'fire biome' represented more than 50% of the observed AF pixels in the fire were retained for use in $EC_{CO}^b$ derivation.

For each fire in the final match-up set we calculated i) the total CO [g] contained in the plume from the S5P TCCO retrievals, and ii) the total FRE [MJ] released by the fire from Meteosat FRP-PIXEL data – integrating FRP from the time of the fires first AF pixel detection on that day to the moment of the S5P overpass. The minimum TCCO value (mol.m$^{-2}$) from each plume buffer was taken as the appropriate CO 'background' value for that plume and subtracted from all plume pixel TCCO values. The resulting 'excess' TCCO pixel values were then converted to units of grammes by multiplying by the molecular weight of CO and by the pixels' area calculated from their geographic corner coordinates (thus accounting for the change in the along-track spatial resolution of S5P data in 2019 and any view-angle dependent pixel area growth). Summing these values across all plume pixels of a fire then yielded the total amount of fire-emitted CO in that plume, which was compared to the total amount of FRE the fire generated over the time that it released that CO. Each match-up fire FRE and total CO pair constitutes

one datapoint on the relevant 'fire biome' scatterplot of **Figure 2**. FRE and Total CO uncertainties were calculated from the uncertainty values provided within the FRP-PIXEL product and the TCCO product respectively, these are also plotted in the form of error bars on **Figure 2.** In this work the choice was made to not to apply Fixed Mask De-striping (FMD) to the S5P TCCO dataset used as is proposed by Borsdorff et al. (2019), this methodological choice is discussed in more detail in Appendix A.

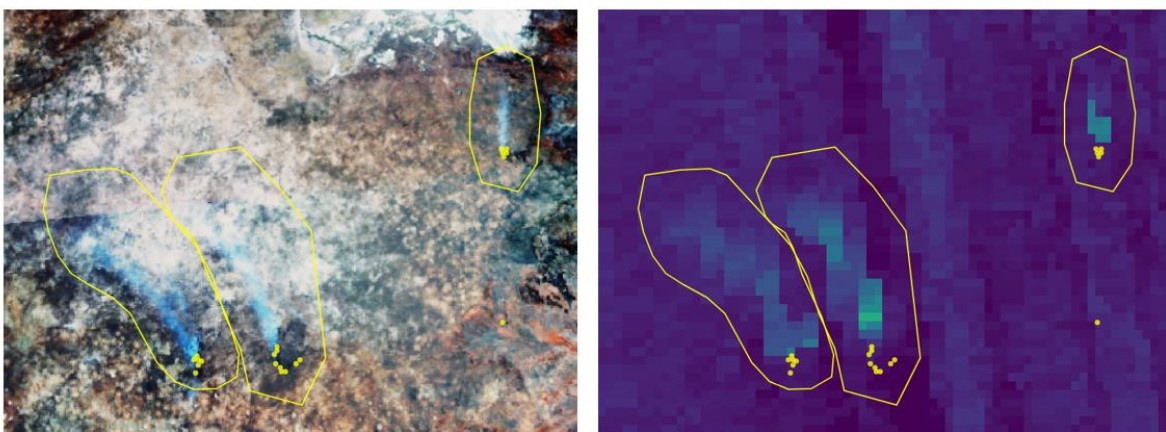

**Figure 1. Example data of a matchup fires used to develop the $C_e^{CO}$ smoke emissions coefficients presented in Figure 2. (a) VIIRS RGB image of three landscape fire smoke plumes, along with (b) the corresponding image of Sentinel-5P derived total column CO (TCCO). In both images, the AF pixel detections taken from the Meteosat FRP-PIXEL product are superimposed (yellow points) along with the bounding polygons used to delineate the fire plumes. Satellites data are from 11:24 and 11:30 UTC respectively, on September 9th over an area in norther Botswana.**

## 2.3    Derivation of Carbon Monoxide Smoke Emission Coefficients $[EC_{CO}^b]$

Matchup data for each 'fire biome' are shown in **Figure 2** and were used to derive the set of biome-dependent CO smoke emission coefficients $[EC_{CO}^b]$ listed in Table 1. Zero-intercept ordinary least squares (OLS) regression was used for this, rather than the orthogonal distance regression (ODR) used by Nguyen and Wooster (2020) during derivation of TPM smoke emission coefficients $[EC_{TPM}^b]$. OLS was used in this work for two main reasons. Firstly, although the ODR method considers the uncertainty in each of the variables, these uncertainties are themselves rather poorly constrained, with the known uncertainties only representing part of the total uncertainty sources. There are contributions to the uncertainty of FRP that are not quantifiable, for example due to variations in the amount of interception of a fire's FRP signal by any overlying tree canopy. We therefore deemed use of a regression method in which the slope is strongly driven by datapoint uncertainty to be unsuitable for use. Secondly, weighting based on uncertainty often resulted in undue weight being given to high value points (e.g. match-ups with high FRE and high plume-species amounts) due to them typically having lower relative uncertainties (see Wooster et al., 2015). Due to their typically being very few high value datapoints in each 'fire biome' due to the heavy tailed nature of

fire size distribution (Freeborn et al., 2009), these few large fires were potentially being too strongly weighted in the resulting calculation of $EC_x^b$. For these reasons, we opted to use OLS regression, and to ensure a consistent methodology for emission coefficient derivation we also applied the same approach to the Nguyen and Wooster (2020) dataset to re-derive their $EC_{TPM}^b$ values using OLS regression (see Appendix B). The updated $EC_{TPM}^b$ for closed canopy forest, managed land, grassland, shrubland, low-woodland savanna and high-woodland savanna are 26.07 g.MJ$^{-1}$, 12.23 g.MJ$^{-1}$, 9.39 g.MJ$^{-1}$, 9.88

g.MJ$^{-1}$, 10.65 g.MJ$^{-1}$, and 14.18 g.MJ$^{-1}$ respectively. On average these new values are 14% lower than those reported in Nguyen and Wooster (2020) derived via ODR, and are the ones referred to and used hereafter. The WRF-CMAQ model-based approach to evaluating our final CO emissions rates and totals in Section 4 was also used to carry out an analogous evaluation of the TPM emissions generated from the updated $EC_{TPM}^b$ values of Appendix B (see Appendix D).

For each of the six fire biomes at least 12 matchup fires were identified for derivation of $EC_{CO}^b$, apart from for closed canopy forest. Tropical evergreen forests (the primary type of closed canopy forests in tropical regions) are generally not very susceptible to fire, except during periods of extreme drought, due to the high humidity and low windspeed within the dense forest canopy and the limited amount of surface fuel available due to rapid decomposition of surface litter in these environments (Marengo et al., 2011; Tomasella et al., 2013). Fires in such tropical forests are most often the result of human land-clearing

activity and are typically small in size, unless heavy machinery is involved in land clearing (Van Leeuwen et al., 2014). Furthermore, FRP observations in closed canopy forest can be affected by tree canopy interception of surface emitted FRP (Roberts et al., 2018). These factors result in a lower number of observable and identifiable fire match-ups for tropical closed canopy forest areas. Smaller fire sizes and fewer match-up fires being acquired in closed canopy forests areas relative to other biomes was also observed during previous applications of the FREM approach (Mota & Wooster, 2018; Nguyen & Wooster

2020) and the FEER approach (Ichoku & Ellison, 2014), even when using 7 years' worth of MODIS FRP and AOD data in the latter case. In this work the ability to identify small fires in closed canopy forest is further limited by i) the spatial resolution of the S5P TCCO observations which are at least 5 times lower resolution than the 1 km AOD product used in Nguyen & Wooster (2020) and ii) the limited availability of the S5P trace gas products which only became operational form mid-2018. An increased timeseries of S5P data and the exploitation of machine learning methods such as object recognition may aid in

identifying a greater number of plumes in closed canopy forest - and this the subject of ongoing work.

Due to the FEER emission inventory exploiting a far larger dataset from which to identify fire match-ups (7 years of MODIS FRP and AOD) and it obtaining many more fire match-ups in tropical closed canopy forest (Ichoku and Ellison, 2014) we instead derive $EC_{CO}^b$ for closed canopy forest from the 'FEER-equivalent' value. The method used to derive this is detailed in

Nguyen and Wooster (2020), and essentially involves aggregating the FEER $C_e^{TPM}$ emission coefficients of Ichoku and Ellison (2014) (https://feer.gsfc.nasa.gov/data/emissions/) to the relevant fire biome. Equation 1 was then applied to obtain a FEER-equivalent $EC_{CO}^b$ , which was calculated as 156.7 g.MJ$^{-1}$ for the closed canopy forest fire biome. We generated FEER-

equivalent $EC_{CO}^b$ for each of the other five fire biomes to compare these to our directly derived $EC_{CO}^b$ values, and found agreement within ±34% (see Table 1), somewhat justifying our use of the FEER-equivalent value in the closed canopy forest biome where a directly derived $EC_{CO}^b$ value was not achieved. Further, Nguyen and Wooster (2020) showed that mean monthly FRE contribution from fires in closed canopy forests does not exceed 10% of the total monthly FRE coming from African fires, and thus its total is of relatively low importance to continental-scale CO emissions totals.

Tests were carried out to determine the statistical significance of the $EC_{CO}^b$ values derived in Figure 2. Table 2 details the resulting p-values and t-values for each pair of biomes and shows that only one pair of biomes (grassland and managed land) have $EC_{CO}^b$ values which are statistically different at the 95% confidence limit. Two additional biome pairs are statistically distinct at the 85% confidence interval (grassland-low woodland savanna and managed land-shrubland). This analysis indicates that in general, based on the current match-up dataset, different biomes do not have statistically significant $EC_{CO}^b$ values i.e. the type of vegetation being burned does not result in a statistically different mass of CO being observed. However, from what is known about emission factors (e.g. Akagi et al., 2011; Andreae, 2019) this is physically unrealistic. The updated FREM $EC_{TPM}^b$ values of Appendix B were similarly tested for statistical significance and the emission coefficients generated from this much larger match-up dataset (primarily due to the higher spatial resolution of the MCD19A1 AOD product allowing more plumes, particularly smaller plumes, to be identified) were overall more statistically different. Of the fifteen biome pairs only four were not statistically significant at a 95% confidence threshold and this reduced to 2 pairs at the 85% confidence limit. The fire-plume matchups used in $EC_{CO}^b$ (and $EC_{TPM}^b$) derivation are classified into one of six distinct 'fire biome' classes based on the fractional contribution of active fires in these different biomes to the fire. As such, in most cases a smoke plume is not 100% generated by fire that burns vegetation of a single biome. As detailed in Section 2.2, a filter is applied to ensure that the majority (>50%) of the active fire pixels responsible for a smoke plume come from a single biome, and only if this condition is met is the fire-plume matchup included in the $EC_{CO}^b$ derivation for that biome. A larger plume dataset would allow the application of a more stringent classification condition (e.g. > 70% of active fire pixels being from a single biome) whilst still maintaining a reasonable matchup sample size to generate $EC_{CO}^b$ values from for each biome. Having an increased number of matchup fire-plumes which have higher fractional coverage of one single biome would likely result in $EC_{CO}^b$ values that are more statistically distinct from one another. In the derivation of $EC_{TPM}^b$ described in Appendix B, the 1 km spatial resolution of the AOD product used allowed the inclusion of more small fire-plumes to generate $EC_{TPM}^b$ values. These smaller fire-plumes were generated by fires covering smaller areas, meaning that there were more fires included in the matchup dataset which had a larger majority of their AF pixels coming from a single biome. Indeed, resulting $EC_{TPM}^b$ values generated by this dataset were more statistically distinct from one another (See Appendix B) than the $EC_{CO}^b$ values herein The barriers to producing a larger sample size in this study have previously been discussed along with the future research focus needed to address this key issue.

For ease of future discussion, hereafter, we will refer to emissions inventories generated using the $EC_{CO}^b$ coefficients of this Section as the FREM_bCO emissions inventory or FREMs_bCO when the small fire correction is applied. Any emissions inventory generated using the updated $EC_{TPM}^b$ coefficients reported in **Table 1** and detailed in Appendix B will be referred to as FREM_bTPM hereafter. The *b* in both cases denotes the *base* or reference species used to produce emissions estimates.

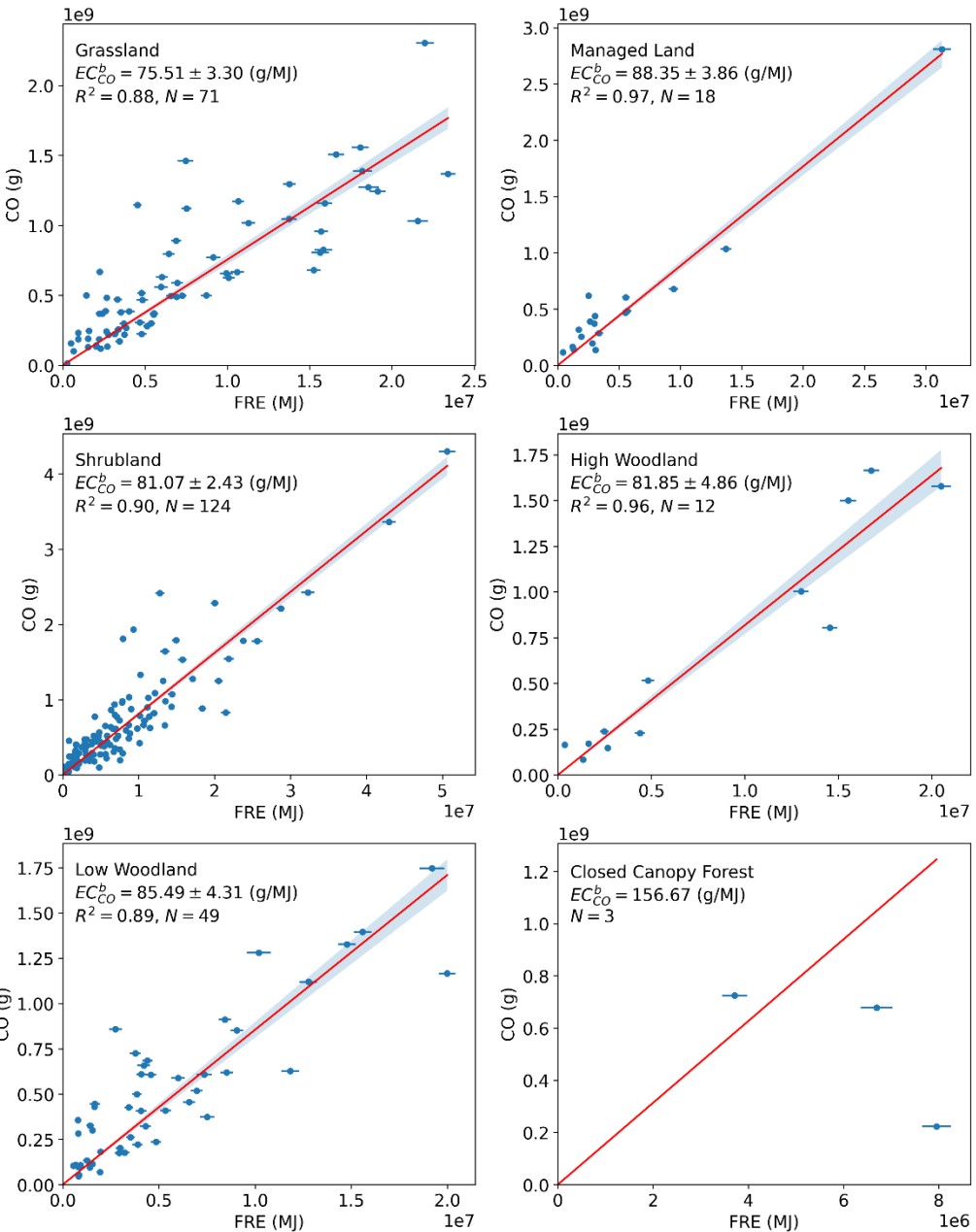

**Figure 2. Carbon monoxide smoke emission coefficients ($EC_{CO}^b$; in g.MJ-1) derived from matchup fires burning across the six fire-affected biomes shown mapped in Figure 8b across southern hemisphere Africa (note that the matchup fires here come from both**

African hemispheres). Each $EC_{CO}^b$ is derived from the slope of an ordinary least squares (OLS) regression between the fire-emitted CO calculated from Sentinel-5P total column CO (TCCO) observations and the fire's matching FRE. The shaded grey area indicates the error of each slope. Error bars represent the uncertainty in both the FRE and total plume CO calculated from the uncertainty measures of the FRP-PIXEL product and the S5P TCCO product respectively. Closed canopy forest had insufficient matchup fires identified and so $EC_{CO}^b$ for this biome was derived using the FEER-equivalent procedure detailed in Section 2.3 and in Nguyen and Wooster (2020). Datapoints from the three matchup fires that were identified in closed canopy forest are included in the plot.

Table 1. Biome-dependent CO smoke emission coefficients ($EC_{CO}^b$ $in$ $g. MJ^{-1}$) as derived from the data shown in Figure 2. Also shown are the matching values calculated using the updated FREM $EC_{TPM}^b$ values from Appendix B and FEER-equivalent coefficient values produced from the FEER product of Ichoku & Ellison (2014), aggregated to the FREM biomes (see Nguyen and Wooster (2020) for full details). $EC_{CO}^b$ values calculated via the updated FREM $EC_{TPM}^b$ values shown in Appendix B are included to demonstrate the impact of using a different reference species and the associated EFs in Equation 1 to estimate $E_{CO}^b$. *Figure 2 shows insufficient matchup fires were found for the closed canopy forest biome, so the FEER-equivalent value is reported instead and used hereafter.

| Fire Affected Biome | Sentinel-5P TCCO-derived $EC_{CO}^b$ (Section 2.2) | $EC_{CO}^b$ calculated via updated FREM $EC_{TPM}^b$ (see Appendix B) | FEER-equivalent (see Nguyen & Wooster, 2020) |
| --- | --- | --- | --- |
| Closed Canopy Forest | 156.7* | 248.7 | 156.7 |
| Managed Land | 88.4 | 72.1 | 100.7 |
| Grassland | 75.5 | 74.5 | 87.5 |
| Shrubland | 81.1 | 78.4 | 87.4 |
| Low-woodland savanna | 85.5 | 84.5 | 101.9 |
| High-woodland savanna | 81.9 | 112.5 | 110.1 |

**Table 2. Results detailing the statistical significance of the $EC_{CO}^{b}$ values derived for each biome pair. P-values below 0.05 are coloured green, those below 0.15 are coloured yellow, while any biome pair with p-values above 0.15 are coloured red.**

| Biome 1 | Biome 2 | p-value | t-value |
|---|---|---|---|
| grassland | managed land | 0.013 | 2.53 |
| grassland | shrubland | 0.176 | 1.36 |
| grassland | high woodland savanna | 0.283 | 1.08 |
| grassland | low woodland savanna | 0.068 | 1.84 |
| managed land | shrubland | 0.112 | 1.60 |
| managed land | high woodland savanna | 0.304 | 1.05 |
| managed land | low woodland savanna | 0.622 | 0.49 |
| shrubland | low woodland savanna | 0.373 | 0.89 |
| shrubland | high woodland savanna | 0.886 | 0.14 |
| high woodland savanna | low woodland savanna | 0.577 | 0.56 |


## 3    FREM Fire Emission Inventory

### 3.1    CO Emissions

Following derivation of a $EC_{CO}^{b}$ for each of the six fire biomes of NHAF and SHAF (Table 1), a set of landscape fire emission rates and totals for these regions were derived via application of $EC_{CO}^{b}$ values to the complete Meteosat FRP-PIXEL data record of 2004 to 2019 (Wooster et al., 2015). To account for possible changes in landcover, fire biome maps for 2005, 2010 and 2015 were produced in an analogous method to that used to generate the 2019 map (i.e., based on the CCI Landcover and Landsat VCF data products of these years). The Meteosat FRP-PIXEL record was combined with this set of biome maps and the $EC_{CO}^{b}$ values of Table 1 to produce a 16 year record of African fire emissions. This FREMs_bCO inventory ('s' denoting the application of the small fire correction) is the highest spatio-temporal resolution fire emissions inventory for CO yet available over Africa (15 min, 3 km at the sub-satellite point). Monthly totals of FREMs_bCO emissions with cloud and small fire correction applied are shown in Figure 3 alongside those of the most current version of the Global Fire Emissions Database (GFEDv4.1s; van der Werf et al., 2017; www.globalfiredata.org/). GFED4.1s includes its own 'small fire' correction to account for bur scars undetected in the 500 m MODIS MCD64A1 burned area product. Mean annual CO emission totals are detailed for both inventories with and without their respective small fire corrections applied in Table 33.

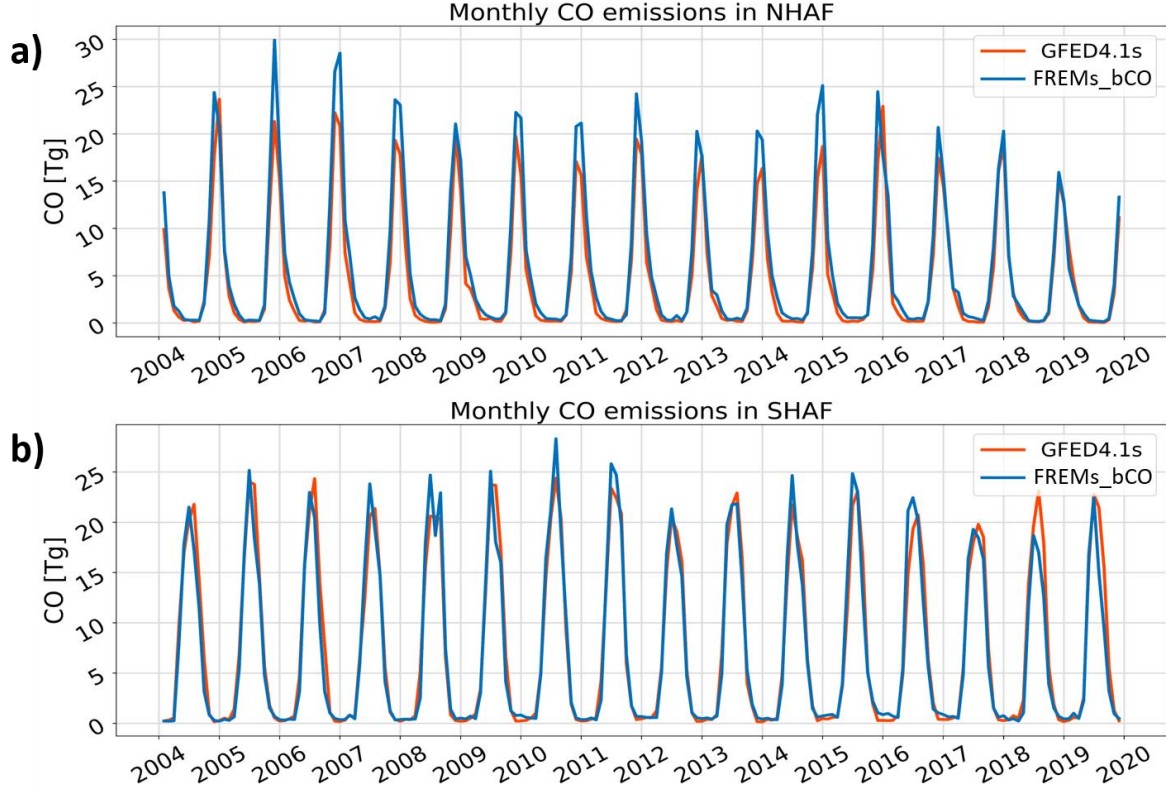


**Figure 3. Monthly landscape fire CO emissions over a 16-year period for (a) northern and (b) southern hemisphere Africa, as derived in FREMs_bCO and GFED4.1s, both with their respective 'small fire correction' applied ("s" indicates its application)**

**Table 3. Mean annual CO fire emission totals for the period 2004 to 2019 as derived for northern and southern hemisphere Africa using the FREM methodology developed herein and reported alongside those of GFEDv4.1. Values are reported as those both with and without the relevant 'small fire correction' applied ("s" indicates its application), as well as the % difference made by this upward adjustment.**

| | | NHAF Total CO Emissions [Tg] | SHAF Total CO Emissions [Tg] |
|---|---|---|---|
| Without small fire correction applied | FREM_bCO | 46.6 | 64.4 |
| | GFED4.1 | 40.5 | 67.5 |
| | | | |
| With small fire correction applied | FREMs_bCO | 70.1 | 87.2 |
| | GFED4.1s | 55.9 | 90.2 |
| | | | |
| FREM_bCO | % effect of small fire correction | 50.4 | 35.4 |
| GFED4.1 | % effect of small fire correction | 38.0 | 36.7 |

The FREMs_bCO and GFEDv4.1s CO emissions time-series shown in Figure 3 show very similar magnitudes, particularly in SHAF. Table 3 confirms that the mean annual totals are also close, with FREMs_bCO 25% higher than that of GFED4.1s in NHAF, and 3% lower in SHAF. The small fire corrections of both inventories also increase the basic CO emissions calculated in each inventory by similar a magnitude, especially in SHAF. The closeness of these results is noteworthy when considering that these CO emissions estimates have been produced using completely different methodologies and with no input data,

conversion variables or emissions factors in common. Figure 3 also shows similar annual temporal patterns between the two inventories, with annual peaks and minima generally occurring in the same years. However, as Mota and Wooster (2018) and Nguyen and Wooster (2020) noted for TPM emissions, the FREM methodology often predicts a slightly earlier peak in annual emissions in SHAF compared to GFED. This shifted peak agrees with findings showing that polar-orbiting based FRP observations also seem to peak in SHAF a month or so earlier than do BA observations, for example, in the work of Zheng

et al., (2018) who compare GFED BA with GFAS FRP. The same work also suggests that measured CO emissions actually lag BA derived CO emissions in Africa, based on MOPITT CO observations. Zheng et al. (2018) attribute this lag in BA-based CO emissions and observed CO emissions to a shift from flaming to smoldering combustion over the continent which is not accounted in the emissions factors applied in the GFED procedure..

A more spatially detailed intercomparison is shown in Figure 4 examining a month of FREMs_bCO hourly average CO emissions in two of the most fire affected countries in Africa - the Central African Republic (CAR) and Angola during January

and August 2012 respectively (typically their peak fire months). GFED hourly averages were calculated by dividing GFED4.1s monthly emissions totals by the number of hours in each month. Our mean hourly CO emissions for CAR in January are lower than those of GFED4.1s by 40%, whereas for Angola in August they are 60% higher. The very strong fire emissions diurnal cycle is highly resolved by the FREM inventory, demonstrating the data richness provided by the high temporal resolution of the geostationary FRP observations used. An additional benefit is that, unlike burned area data, FRP observations from geostationary satellites are available in near real-time and thus the FREM emissions of CO, TPM and other air pollutants are a potential source of data for air quality forecasting (Roberts et al., 2015).

The small differences seen between the FREMs_bCO and GFEDv4.1s CO emissions at the hemisphere scale (Figure 3) compared to the larger country-level differences (Figure 4) demonstrate how emissions inventories may be similar in magnitude at larger scales but can vary significantly more at the local scale. Zhang et al. (2014) compared modelled AOD fields generated from seven commonly used fire emissions inventories using an atmospheric transport model and demonstrated that the maximum variation between the modelled AOD averages of these inventories increased significantly when moving from regional to local scale in Northern Sub-Saharan Africa.

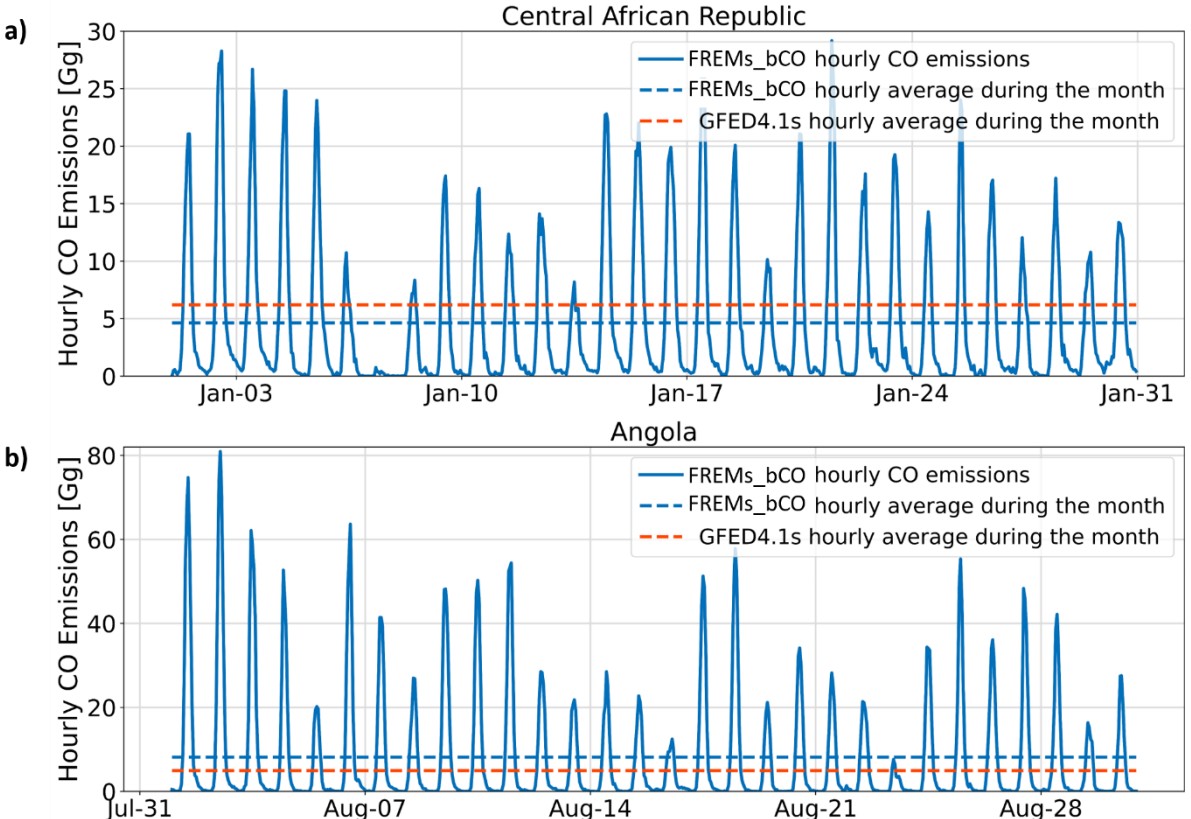

**Figure 4. Hourly FREMs_bCO emissions from landscape fires burning over a month-long period in 2012 during the peak fire season of (a) Central African Republic and (b) Angola. The monthly mean of this emission rate is also shown, along with that from**
**GFEDv4.1s.**

Past comparisons made between modelled CO atmospheric concentrations driven by GFED, and CO observations coming from instruments such as MOPITT (Worden et al., 2010) suggest that GFEDv3 underestimated African CO emissions by up to 50% (Chevallier et al., 2009; Kopacz et al., 2010; Pechony et al., 2013). Since GFEDv3 and GFEDv4.1s CO emissions are
similar both for NHAF and SHAF, this points to a possible continued underestimation of CO emissions by GFEDv4.1s over Africa. Each GFED version uses the 500 m MODIS MCD64A1 burned area product as their driving data, and recent studies have shown African burned area to be higher than MODIS estimates when mapped using 20 m Sentinel-2 MSI or 30 m Landsat imagery (Hawbaker et al., 2017; Roteta et al., 2019; Tsela et al., 2010). This underestimation by the MODIS BA product is the theoretical basis for requiring the 'small fire correction' in GFEDv4.1 (Randerson et al., 2012; van der Werf et al., 2017).
However, the relatively good agreement seen between GFEDv4.1s and the FREMs_bCO inventory compared herein (e.g. Figure 3) - which are developed from completely different datasets and approaches – could possibly suggest African CO emissions are not so underestimated as past CO observations have suggested when small fires are accounted for. Reconciling top-down and bottom-up derived CO fire emission inventories with observations of CO made from low-earth orbit remains a continuing research focus.


## 3.2    Dry Matter Consumed

Unlike with bottom-up approaches, where DMC [kg] is calculated first and converted to species emissions estimates using estimates of fuel load, combustion completeness and species emissions factors (see Section 1), within the FREM approach fire emissions are estimated directly and DMC can then be calculated from these if required. In this case, DMC is estimated by
dividing the emissions total by the species emissions factor, an approach first demonstrated by Mota and Wooster (2018) using TPM as the relevant species. CO is the second greatest emitted product from biomass burning, and the emissions factor of CO is more consistent and well constrained than that of TPM (Akagi et al., 2011; Andreae, 2019). Therefore, the FREM-derived CO emissions detailed in Section 0 can be related to DMC far more confidently and more consistently than those of TPM. Monthly FREM-derived DMC emissions generated from this approach for CAR and Angola are shown in Figure 5 alongside
those from GFED4.1s. The former are lower at the peak of the CAR fire season compared with those of GFEDv4.1s, but consistently higher at the Angolan fire season peak. Either side of these peaks there is very good agreement between the two.

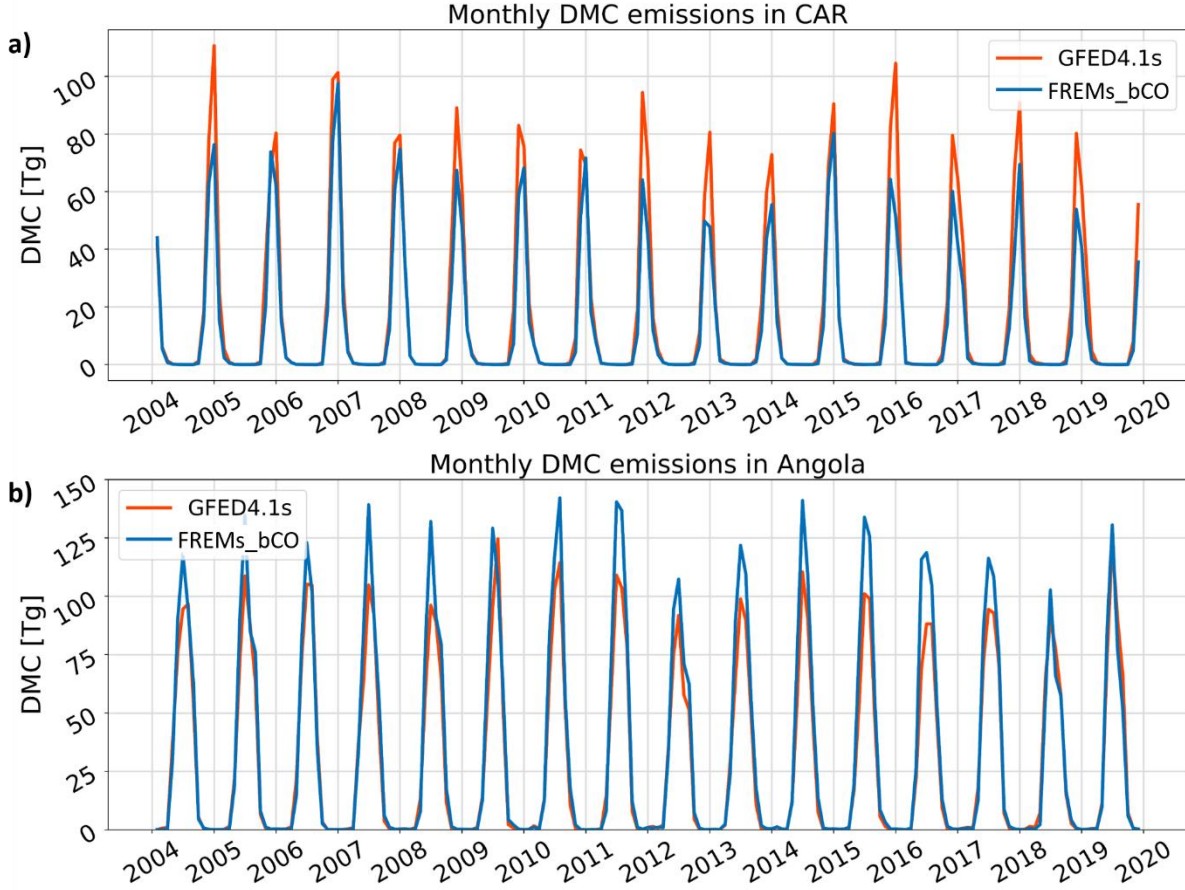

**Figure 5 Monthly dry matter consumed (DMC in g) for a 16-year period as derived from FREMs_bCO CO emissions and that of GFED4.1s for (a) Central African Republic (CAR) and (b) Angola.**


Once calculated, DMC can be further combined with burned area information to generate DMC per unit burned area measures across the African region – the only observation based approach capable of doing this at present (Nguyen and Wooster, 2020). We use FREM-derived DMC with the Sentinel-2 20 m spatial resolution FireCCI Small Fire Dataset (v2.0) for 2019 to calculate DMC per unit area at 0.1° resolution in that year. These observation-based DMC per unit burned area values are

shown in Figure 6, alongside BA based values reported in GFED4.1s for 2019 at 0.25° resolution. Total carbon emissions can be easily calculated using the assumed carbon fraction of vegetation (taken typically as 50±5%) (Andreae, 2019). Focusing in on a 4° × 4° region of Zambia (Figure 7) demonstrates the higher spatial detail of the FREM-derived DMC per unit burned area data compared with that provided by the modelling used within GFED.

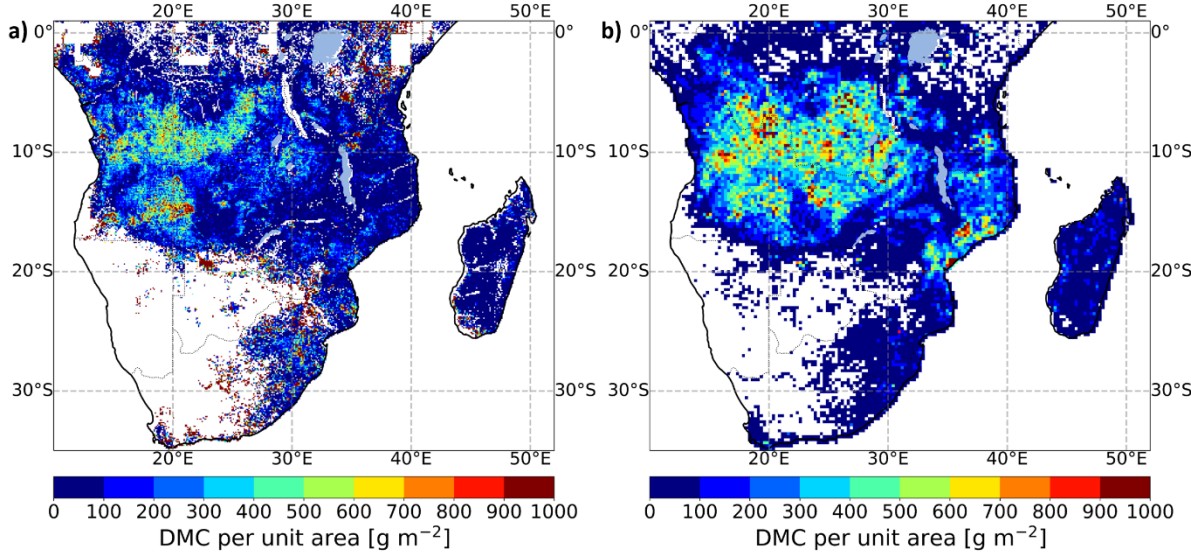


**Figure 6 Dry matter consumed (DMC) per unit area, mapped across southern hemisphere Africa for 2019. (a) as calculated in 0.1°
grid cell resolution by dividing the FREMs_bCO CO values (with SF correction) by the 20m FireCCISFD burned area product
generated from Sentinel-2 MSI observations, and (b) as reported in GFED4.1s at 0.25° grid cell resolution.**


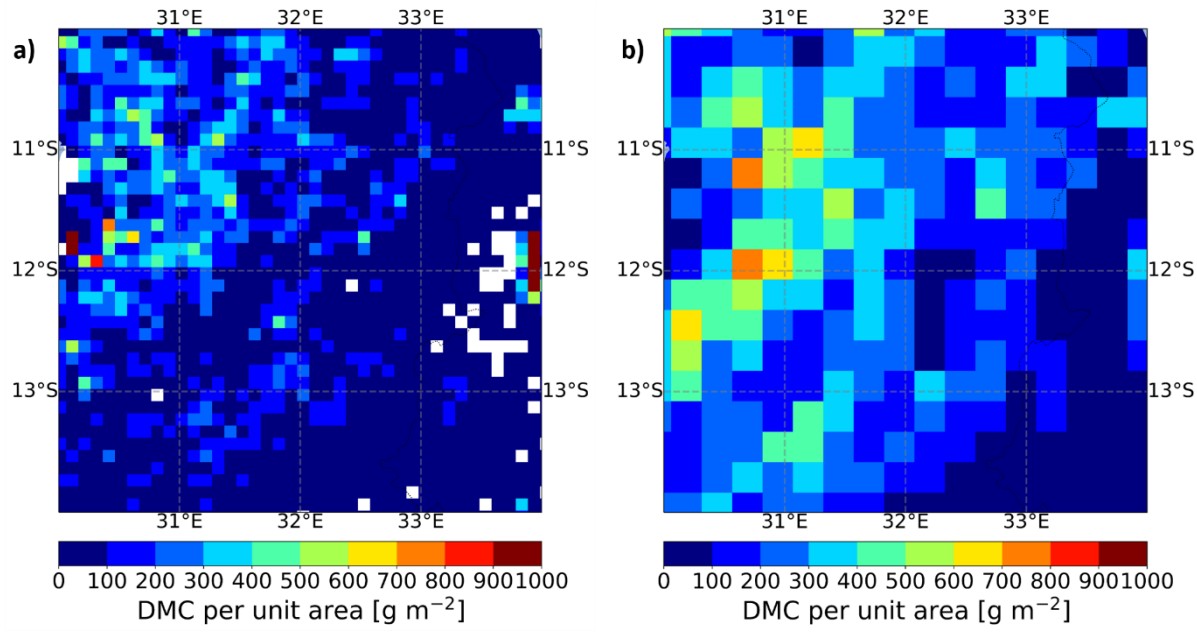

**Figure 7. Mapped dry matter consumed (DMC) per unit area, calculated (a) at a 0.1° grid cell scale using the FREMs_bCO CO
values (with SF correction) across a 4° × 4° region of Zambia for the year 2019 and (b) the same DMC per area values for GFED4.1s.**

## 4 FREM-Derived CO Emissions Assesment

### 4.1 Evaluation Methodology

Beyond the comparisons to GFED4.1s detailed in Section 3, our FREMs_bCO emissions were further evaluated through their use in chemical transport model (CTM) simulations conducted with the Advanced Research Weather Research and Forecasting model (WRF-ARW v4.1.1; (Skamarock et al., 2019) and the Community Multiscale Air Quality model (CMAQ v5.3; EPA, (2019); https://www.epa.gov/cmaq). The resulting model output fields (that used FREMs_bCO emissions as input) were compared to Sentinel-5P TROPOMI TCCO observations that were completely independent of those used in the FREM emissions coefficient generation (i.e. those used within Figure 2).

WRF-CMAQ is commonly used in operational AQ systems (Kukkonen et al., 2012) and in research related to fire emissions and smoke-contaminated air (Cheng et al., 2014; Baldassarre et al., 2015; Hu et al., 2016; Vongruang et al., 2017; Koplitz et al., 2018; Choi et al., 2019). Model runs were conducted over the ~ 3000 $km^2$ region of SHAF shown in Figure 8a. Further regions of interest (ROIs) were used in comparisons between the WRF-CMAQ output and independent S5P TCCO observations. The WRF-CMAQ domain had a spatial resolution of 9 km, with 35 vertical model layers over a $347 \times 319$ grid. Model runs were conducted for the period 15[th] June to 29[th] August 2019 and were carried out in two separate simulations each initialised and fed with initial and boundary conditions from a global meteorological (FNL; https://rda.ucar.edu/datasets/ds083.2/) and chemistry (WACCM; https://www2.acom.ucar.edu/gcm/waccm) model. The first half of June was excluded due to a change in the version of the global metrological model used as input. The second simulation was started from 29[th] July 2019 and both simulations featured a 24-hour spin-up time. The model configuration and set of physical schemes used in WRF were selected based on previous AQ simulations over SHAF using the WRF-Chem model (Kuik et al., 2015; Yang et al., 2013; Zhang et al., 2014). Details of the WRF-CMAQ configuration and setup are summarised in Appendix C. Anthropogenic emissions were taken from the EDGAR-HTAPv2 inventory (https://edgar.jrc.ec.europa.eu/dataset_htap_v2), whilst biogenic and dust emissions were generated inline by the model. The FREMs_bCO biomass burning inventory was used as input for CO emissions and emission coefficients for all other gas species required by the CMAQ model were calculated through the application of Equation 1 using $EC_{CO}^b$ values. These emission coefficients were then multiplied by hourly mean SEVIRI FRP to generate all the fire-emitted gas and particulate species emissions required as input in the model. Aerosol species emissions were generated through an analogues application of the updated FREM $EC_{TPM}^b$ emission coefficients of Nguyen & Wooster (2020) (see Appendix B)

The CMAQ model produces a TCCO output (mol.m$^{-2}$) that could be compared to Sentinel-5P TCCO (mol.m$^{-2}$) measurement data from June to August 2019. None of the S5P observations used in this comparison were those deployed in the $EC_{CO}^b$ derivation of Section 2.3. Prior to the intercomparison, both model and measurement datasets were converted to units of g.m$^{-2}$ via multiplication by the molecular mass of CO. S5P acquisitions over the model domain occur daily between 12:00 and

14:00 UTC, and the resulting TCCO retrievals were combined and compared with the mean CMAQ TCCO output from the same two-hour period. Both modelled and observed TCCO were mapped to a 0.1° grid and their degree of agreement quantified using the Pearson's correlation coefficient (r) and the normalised mean bias function (NMBF) described by (Yu et al., 2006). The NMBF has been specifically developed for comparing modelled and observed air pollutant concentrations, and it reduces the inflation in bias that may be caused by low values of the observed quantities (see Yu et al., 2006). NMBF is defined as:

$$NMBF = \frac{(\sum M - \sum O)}{|\sum M - \sum O|} \cdot \left[ exp\left( \left| ln \frac{\sum M}{\sum O} \right| \right) - 1 \right] \qquad [2]$$

where $M$ and $O$ are the modelled and observed TCCO. As defined above, a positive NMBF indicates an overestimation of the model by a factor of $1 + $ NMBF, while a negative NMBF indicates that the model underestimates observations by a factor of (1-NMBF). Hence, a NMBF value of 0.10 is a 10% overestimation by the model, and -0.10 a 10% underestimation.

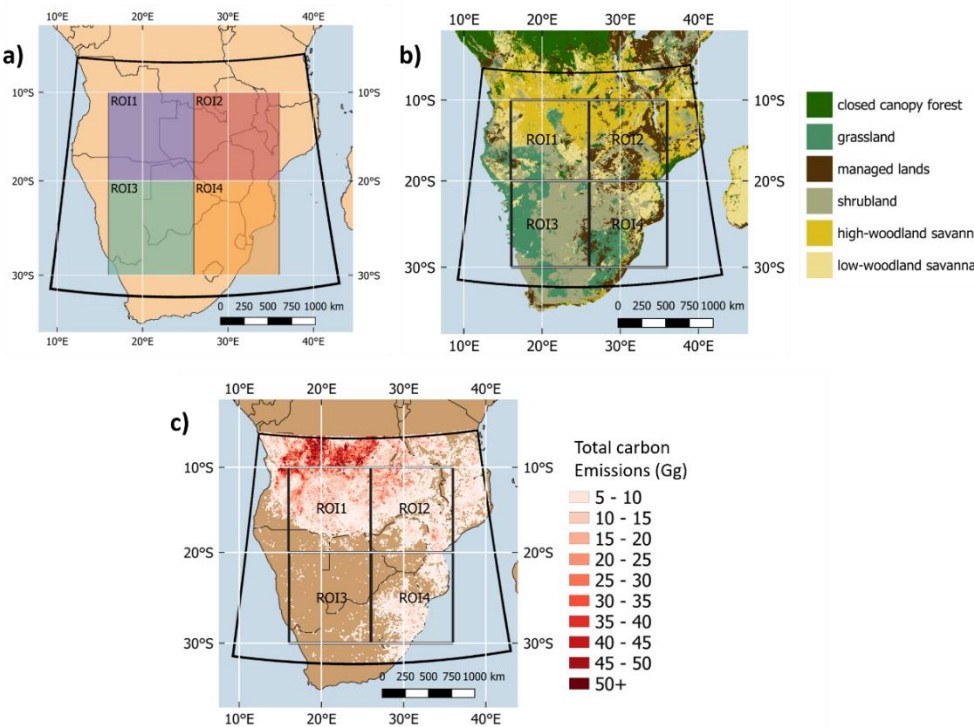

**Figure 8. Southern hemisphere Africa (SHAF) model domain in WRF-CMAQ, fed with the FREM-derived landscape fire CO emissions inventory developed herein. Boxes indicate four smaller regions of interest (ROIs) used in comparisons of model output to satellite-derived CO observations. (a) Domain and ROIs, (b) spatial distribution of the six fire-affected biomes defined herein, and (c) spatial distribution of fire emitted total carbon released between 15th July and 29th August as estimated from the FREMs_bCO emissions inventory, calculated through the application of Equation1 to obtain DMC and then carbon fraction. This FREMs_bCO inventory is used as input to the CMAQ model.**

## 4.2 Evaluation Results

Mapped mean monthly TCCO (g.m$^{-2}$) from the CMAQ model and S5P TCCO observations are shown in Figure 9, along with their percentage difference. In general, their spatial distribution agrees well - with the highest TCCO values in the northwest of the domain – which is the area with greatest fire activity (Figure 8c). The magnitude of TCCO over this region in the CMAQ model output is however higher than that of the S5P observations, by around 50% in some areas in June and July. Across the majority of the rest of the domain however, modelled TCCO is between 1% and 30% lower than observed TCCO. An improved agreement is seen in August, with the degree of over and underestimation of CMAQ compared to S5P generally reduced.

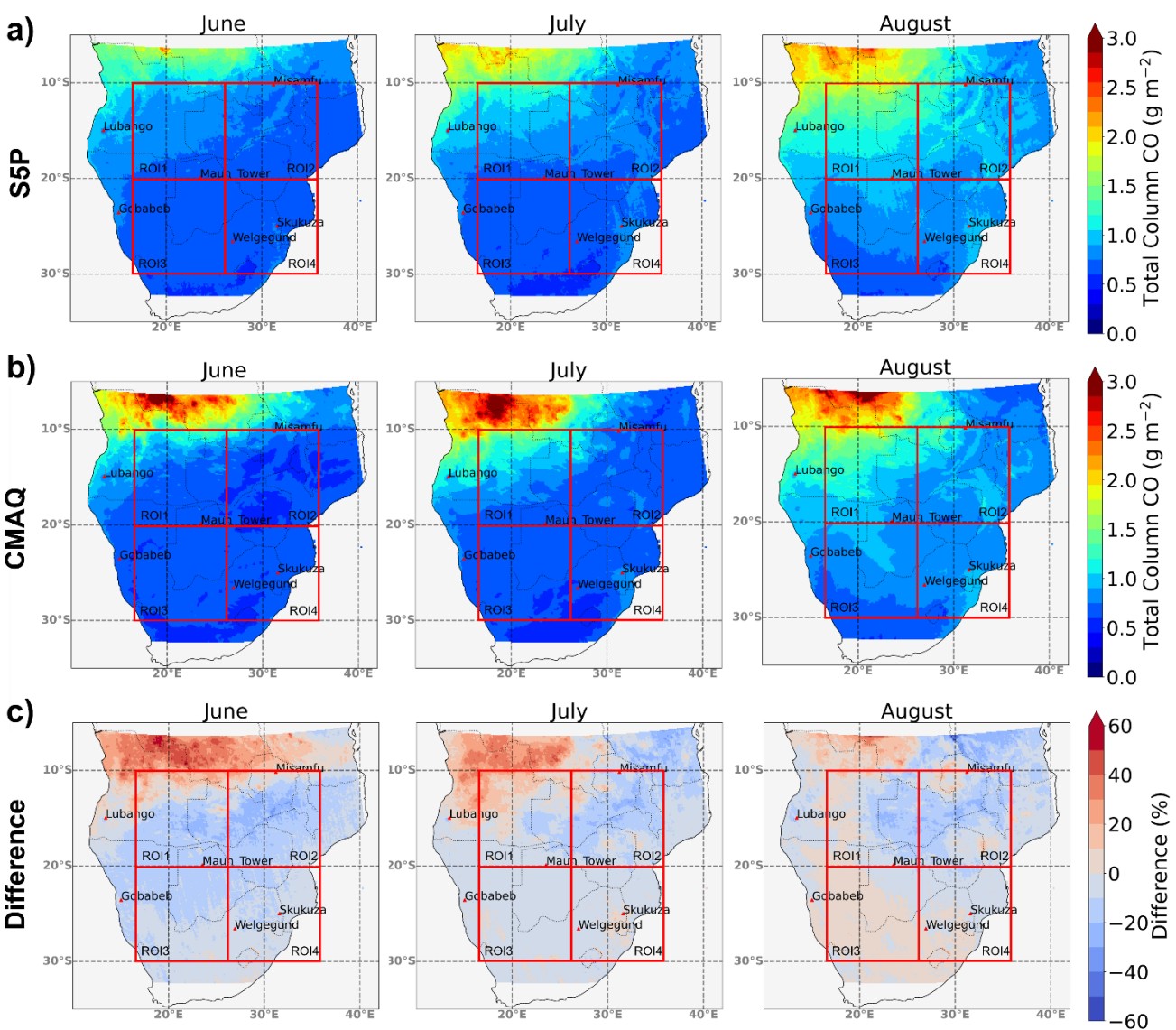

**Figure 9. Mapped mean monthly total column carbon monoxide (TCCO) between 15th June and 29th August 2019, as determined by (a) S5P observations typically observed between 12:00and 14:00 UTC and (b) CMAQ averaged model TCCO output between 12:00 and14:00 UTC, the model is fed with the FREMs_bCO emissions inventory developed herein. Their percentage difference is shown in (c).**

Area-aggregated CO totals (Gg) were calculated by multiplying both CMAQ and S5P TCCO by their $0.1° \times 0.1°$ grid cell

areas to obtain a daily summed total CO timeseries (between 12:00 and 14:00 UTC) for the full domain extent and within

ROI1 and ROI2 (labelled in Figure 8). These are shown in Figure 10, along with direct comparisons of these daily area-

aggregated estimates. These CO totals show that temporal patterns observed by S5P are well replicated by the CMAQ

modelling driven by the FREM-derived CO emissions. This indicates that (i) temporal trends in active fires are being well

captured in the SEVIRI FRP-PIXEL product and (ii) the meteorological fields of WRF, particularly wind, are representing the

490 real conditions sufficiently well. In direct comparisons between daily area-aggregated total CO across the four ROIs (Table

44), ROI1 shows the best agreement between model and observations (NMBF = -0.01; a 1% underestimation by CMAQ

compared to S5P). Daily total-area CO for each ROI in each month of the CMAQ simulation period are summarised in Table

44, along with statistics for the comparisons made within each region and month. In the three other ROIs, NMBF lies between

-0.02 and -0.09 for direct CMAQ-S5P comparisons, and both the full domain and all ROIs show a strong correlation between

495 modelled and observed CO (all $r \geq 0.81$).

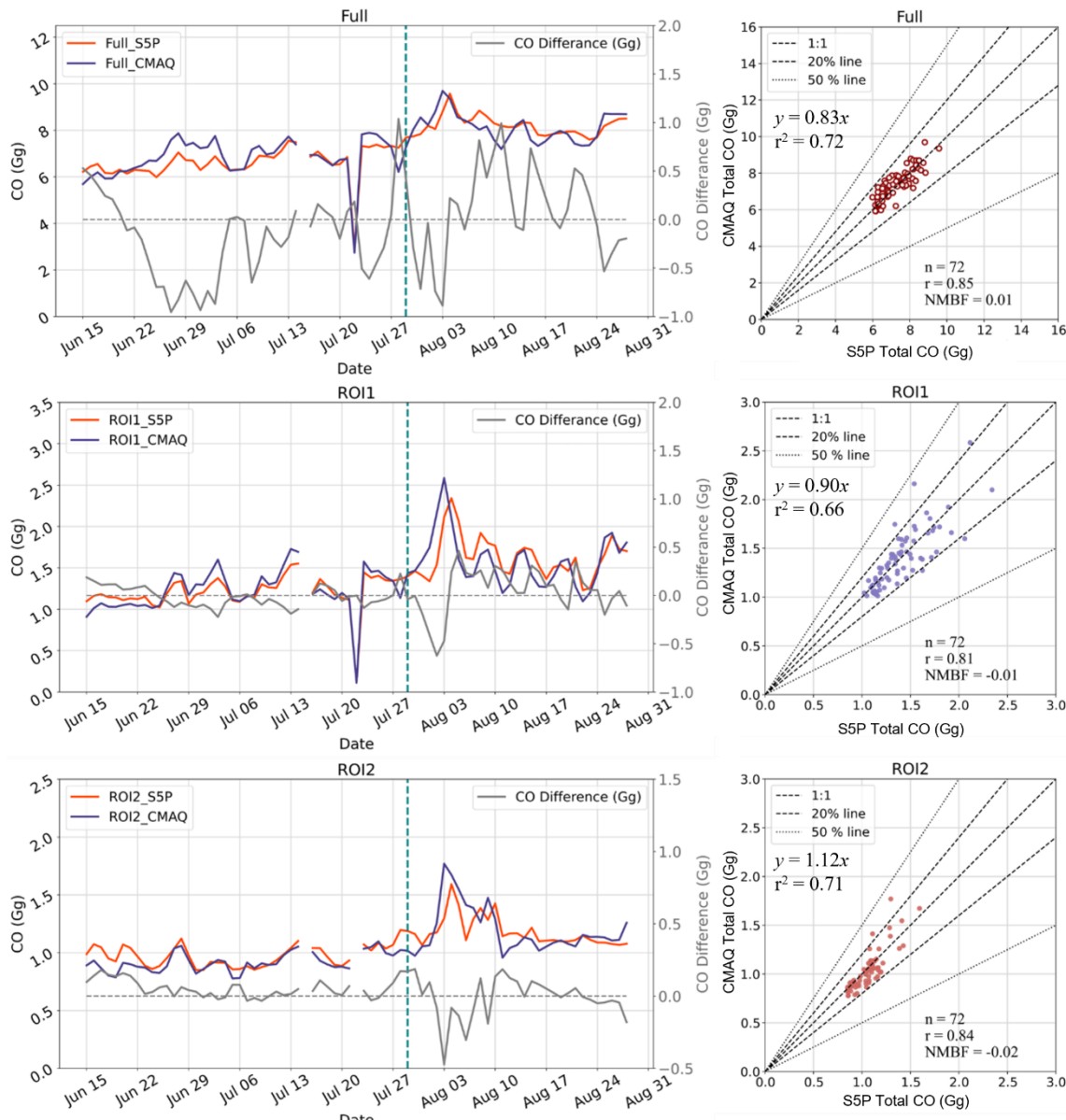

**Figure 10. Results from the comparison of modelled (CMAQ) and observed (S5P)Area-aggregated Total CO within the full domain and two of the regions of interest (ROI) defined in Figure 8. (right-hand column) time series of daily Area-aggregated total CO over the full domain and the two ROIs as determined by CMAQ and S5P, with their difference represented by the solid grey line and the vertical dotted line at 29th July indicating the start of the second simulation (see main text). (left-hand column). Scatterplot comparing the daily Total CO in the right hand column plots from CMAQ and S5P. The Pearson's correlation and NMBF of the dataset are shown, along with dotted lines indicating the 1:1, ±20% and ±50% relationships. The NMBF of 0.04 indicates a mean 4% overestimation by the model compared to the observations. Results from the comparisons plotted here are summarised**

**Table 4. Monthly mean summed TCCO (Gg), as derived across the SHAF domain of Figure 8 from S5P observations and from the CMAQ model output fed with the FREMs_bCO CO emissions. The same values for the four regions of interest (ROI) indicated in Figure 8 are also shown, along with the NMBF and Pearson's correlation coefficient metrics. An NMBF of e.g. 0.05 indicates a mean 5% overestimation of the modelled values compared to the observations.**

|  | CMAQ Mean CO (Gg) | S5P Mean CO (Gg) | NMBF | Pearson's Correlation coefficient |
|---|---|---|---|---|
| **Full Domain** | | | | |
| June | 6.70 | 6.39 | 0.05 | 0.69 |
| July | 7.21 | 7.01 | 0.03 | 0.75 |
| August | 8.12 | 8.24 | -0.01 | 0.70 |
| All | 7.44 | 7.35 | 0.01 | 0.85 |
| **ROI1** | | | | |
| June | 1.14 | 1.16 | -0.01 | 0.79 |
| July | 1.36 | 1.31 | 0.04 | 0.83 |
| August | 1.59 | 1.65 | -0.04 | 0.67 |
| All | 1.40 | 1.41 | -0.01 | 0.81 |
| **ROI2** | | | | |
| June | 0.89 | 0.97 | -0.09 | 0.72 |
| July | 0.95 | 0.99 | -0.04 | 0.83 |
| August | 1.21 | 1.19 | 0.02 | 0.72 |
| All | 1.04 | 1.06 | -0.02 | 0.84 |
| **ROI3** | | | | |
| June | 0.88 | 0.97 | -0.10 | 0.88 |
| July | 1.02 | 1.11 | -0.09 | 0.89 |
| August | 1.24 | 1.33 | -0.08 | 0.61 |
| All | 1.07 | 1.16 | -0.09 | 0.77 |
| **ROI4** | | | | |
| June | 0.73 | 0.78 | -0.08 | 0.84 |
| July | 0.80 | 0.82 | -0.04 | 0.85 |
| August | 0.92 | 0.94 | -0.02 | 0.87 |
| All | 0.83 | 0.86 | -0.03 | 0.90 |

CMAQ-modelled and S5P-observed area-aggregated total CO were also compared for individual smoke plumes. A total of 383 plumes (see Figure 11a) were manually identified through visual inspection of the S5P TCCO product between 15$^{th}$ July and 29$^{th}$ August 2019 and defined using polygons which were then matched to the CMAQ model output at 0.1° resolution. In some cases the spatial distribution of individual plumes in the S5P TCCO product and the CMAQ TCCO output differed slightly – mainly due to differences in the modelled wind direction/speed and the real wind fields. Therefore, a 0.1° buffer was added around each validation plumes' polygon to account for these variations. In the region of highest fire activity (in the north-west region of the model domain) relatively few CO plumes were identified, since the S5P TCCO values were consistently high across this region and individual plumes could not be easily distinguished in the S5P TCCO product. For all identified plumes, in-plume total CO was calculated for both model and observation, again by multiplying TCCO pixels (g.m$^{-2}$) by their grid cell areas and summing these within the bounding polygon containing the plume.

Figure 11b shows the relationship derived between the CMAQ-modelled and S5P-observed in-plume CO. Compared to daily Area-aggregated total CO over the full domain (Figure 10; NMBF = 0.01), in-plume NMBF is slightly higher at 0.04, i.e. a 4% overestimation of the modelled data compared to the observations, while the Pearson's corelation increases from 0.85 to 0.89. The slope of the line-of-best fit to for this data is 1.05 with an r$^2$ of 0.80. Figure 11b shows that the plumes with the highest total CO values (in both the S5P product and in the CMAQ model) also tend to have a higher total CO in CMAQ than in S5P. This is less true for plumes with a total CO below 20 Mg - indicating that the appropriateness of the small fire correction applied, unsurprisingly, depends on the size of the fire i.e. – the FRP contribution from small fires undetected in the SEVIRI product. The time-series of daily mean in-plume CO (Figure 11c) shows that the difference between these measures does not vary significantly by month.

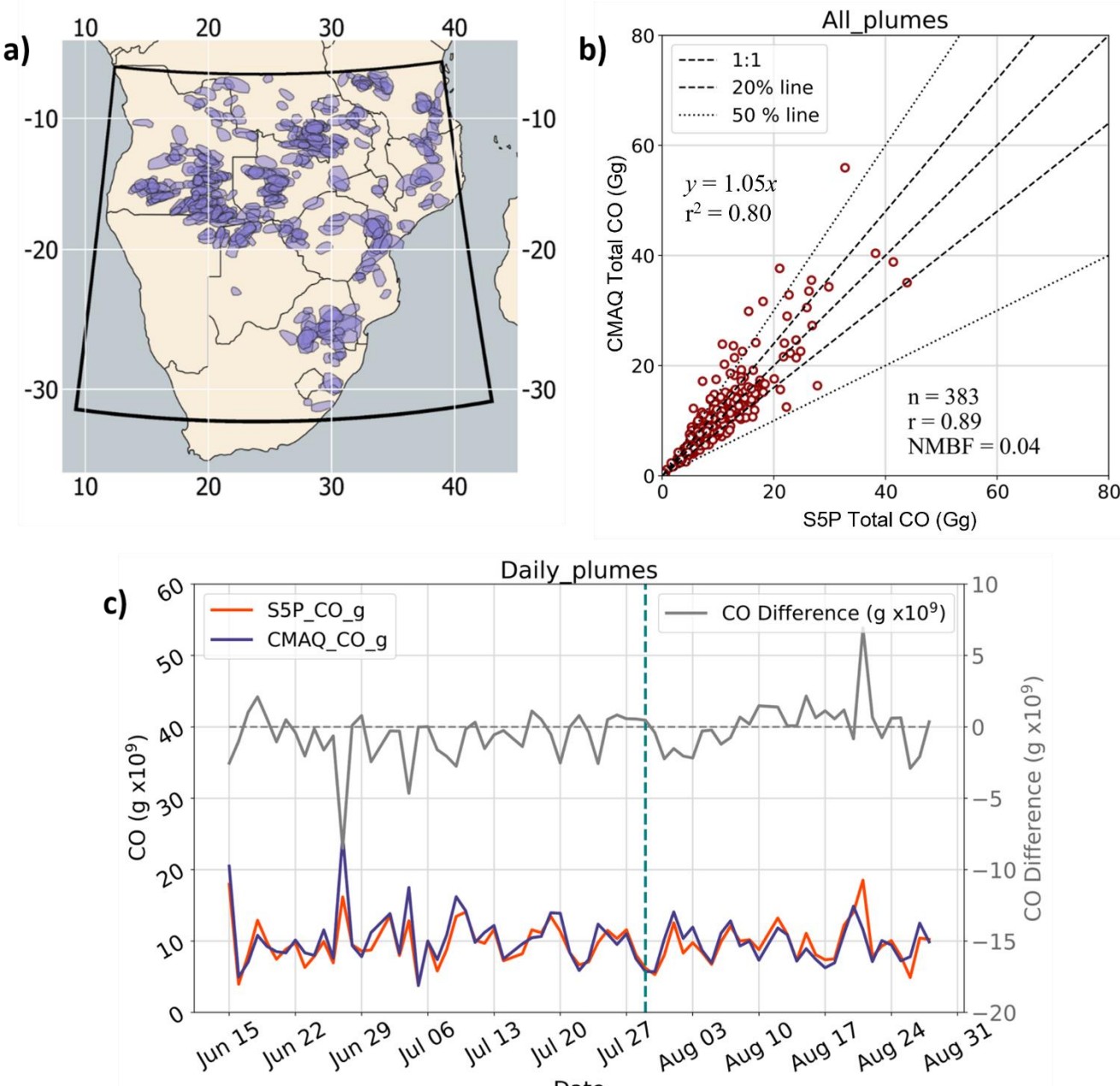

**Figure 11. Evaluation of FREM-derived CO emissions based on WRF-CMAQ modelling. (a) Model domain and the bounding polygons of 383 plumes identified in the S5P TCCO product between 15th June and 29th August 2019 used in the evaluation. (b) Relationship between modelled and observed total in-plume CO ( for the individual smoke plumes identified in (a). The Pearson's correlation and NMBF of the dataset are shown, along with dotted lines indicating the 1:1, ±20% and ±50% relationships. The NMBF of 0.04 indicates a 4% mean overestimation by the model compared to the observations. (c) Summed total CO from all plumes observed on each day of the simulation as determined by CMAQ (purple) and S5P (red), with their difference represented by the grey line (right hand side y-axis). The vertical dotted line on 29th July indicates the start of the second simulation period (see main text).**

## 5 Summary and Conclusions

We have presented significant developments to the 'Fire Radiative Energy Emissions' (FREM) landscape fire emissions methodology of Mota and Wooster (2018) and Nguyen and Wooster (2020), namely the extension to directly relate CO emission rates to FRP observations using an emissions coefficient [$EC_{CO}^b$] derived from satellite total column CO (TCCO) observations and FRE observations. Using 277 matchup fires distributed across northern and southern hemisphere Africa, we have generated $EC_{CO}^b$ values [g.MJ$^{-1}$] for five fire-affected biomes which directly link emission rates of CO (g.s$^{-1}$) to FRP (MW). We have applied these coefficients to the geostationary FRP dataset of African landscape fires from 2004 to 2019 to generate the highest spatio-temporal resolution African CO fire emissions inventory currently available. We find our CO emissions totals to be similar to those of the most recent version of the 'bottom-up' Global Fire Emissions Database (GFEDv4.1s; van der Werf et al., 2017), particularly across SHAF where they are almost identical in magnitude, though featuring a slightly earlier peak in monthly CO emissions coming from FREM compared to GFED.

Since direct validation of large-scale fire emissions estimates remains unfeasible, we have conducted an evaluation of the FREM-derived CO emissions via their use within the WRF-CMAQ atmospheric chemical transport model across a southern African domain. The generated regional-scale total column CO (TCCO) observations are then compared to independent TCCO observations coming from Sentinel-5P TROPOMI. Results of this evaluation indicate very good agreement between the modelled and observed TCCO values in general, and area-aggregated total CO comparisons show a bias of 0.01 and 0.04 (1% and 4% mean overestimation by the model compared to observations) over the full model domain and over individual fire plumes respectively. TCCO emissions are overestimated to a greater extent (by up to around 50%) in the north-west region of the domain where high fire activity is observed, and where CO from fires outside the domain may be being transported into the model domain. The slope of a linear best fit relationship between S5P total CO and CMAQ total CO within individual fire-generated plumes was 1.05 with an r$^2$ of 0.80. In comparison to the ~30% average difference observed between GFEDv3 CO emissions and MOPITT CO observations (Pechony et al., 2013) the results of the evaluation herein show good agreement and are well within the range of biases observed in similar evaluations of other fire emissions inventories (Chevallier et al., 2009; Ichoku and Ellison, 2014; Kaiser et al., 2012; Kopacz et al., 2010; Reddington et al., 2016). The FREM-derived CO emissions produced were used to calculate estimates of Dry Matter Consumed (DMC) and DMC per unit burned area for 2019. The former through use of CO emission factors and the latter through an inversion of the approach of Seiler and Crutzen (1980) in which BA data came from the Sentinel-2 20 m FireCCISFD product of (Roteta et al., 2019). DMC measures produced via FREM-derived CO emissions introduce less uncertainty than those produced from the FREM-TPM emissions of Nguyen and Wooster (2020) (updated in Appendix B) due to CO emission factors being less variable than TPM emission factors in general, especially from fires in tropical forests and cultivated land (Akagi et al., 2011; Andreae, 2019).

Future developments to the approach developed herein will include its application to FRP data from other geostationary satellites, for example those from Himawari (Xu et al., 2017), Meteosat Indian Ocean and GOES (Xu et al., 2010). Emissions of other gases can be derived from the ratio of their emissions factors to those of CO, and this overall approach forms the basis of a new fire emissions product to be delivered by the EUMETSAT Land Surface Analysis Satellite Application Facility (http://landsaf.meteo.pt).

## 6    Appendices

### 6.1    Appendix A

In the work of Borsdorff et al. (2019) a Fixed Mask De-striping (FMD) method is proposed to correct for the striping pattern observed in the operational OFFL S5P TCCO product. This same 'corrected' TCCO is now included as an auxiliary dataset in the operational S5P OFFL CO product (from July 2021). For S5P data prior to this date it is possible to re-create this FMD method and apply it retrospectively to S5P data. We chose not to apply this correction based on our understanding of the FMD method and the 'validation' carried out by Borsdorff et al. (2019), along with observations from our own tests of the impact of the de-striping on smoke plumes.  Primarily, it is currently unclear whether the FMD method is appropriate in areas where very strong spatial CO gradients exist (i.e. within smoke plumes). The evaluation of the FMD method carried out by Borsdorff et al. (2019) involves comparisons of daily averaged de-striped S5P TCCO values against a TCCO measurement made by one of the Total Column Carbon Observation Network (TCCON) ground stations. Whilst the TCCON station was a single point observation, the S5P data were averaged over a colocation radius of 50 km and typically the CO values varied rather mildly across this 50 km region. This is not the case for our fire-generated smoke plumes where changes from 'background' to 'high' column CO occur across very small spatial scales. Further, no TCCON sites exist in Africa so any evaluation carried out could not have been geographically appropriate to our dataset. We therefore chose not to apply the FMD to our S5P TCCO data used in $EC_{CO}^b$ derivation until more quantitative evidence of its appropriateness to such high CO gradient regions is published. For completeness however, we have calculated a set of 'FMD applied' $EC_{CO}^b$ values for comparison, these were generated from the same S5P dataset used in the main $EC_{CO}^b$ derivation of this work, but with the FMD applied. These are shown in Figure A1 and detailed in Table A1. The percentage difference between the $EC_{CO}^b$ values generated from S5P data 'with' and 'without' the FMD applied is also listed.

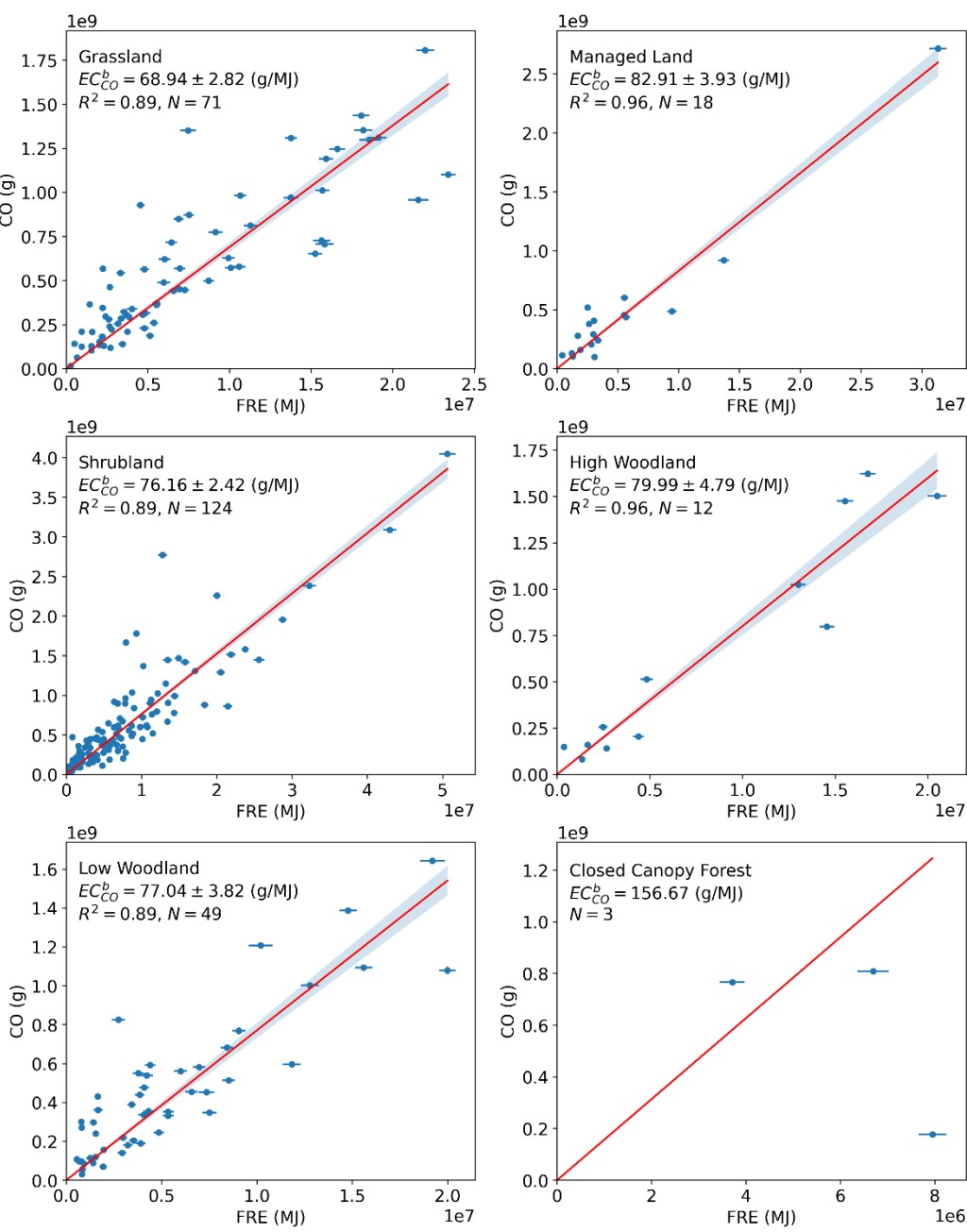


**Figure A1. CO Emissions coefficients [$EC_{CO}^b$] derived with FMD applied to the S5P TCCO data.**

The S5P TCCO product is accompanied by a qa_value for each TCCO pixel which indicates the presence or absence of cloud
in the total column of air that pixel covers. The qa_value assigned determines which averaging kernel is applied in the S5P
TROPOMI retrieval of TCCO  - i.e. the retrieval's sensitivity to CO at different levels of the atmosphere.  A discussion of the
vertical averaging kernels used in the TROPOMI TCCO algorithm can be found in Borsdorff et al. (2014).  It can be seen in
that work that at the typical altitudes at which the large smoke plumes from the types of fires used in our matchup process are
observed (typically 800 m to 4000 m altitude), the weighting of the averaging kernel applied to pixels containing mid-level
cloud (i.e. pixels with qa_values > 0.5 but < 1.0) differs by no more than 0.1 from the weighting of the averaging kernel applied
to clear-sky pixels. In fact, of all altitudes, the sensitivity to TCCO is closest between pixels with qa_values > 0.5 & < 1.0, and
pixels with qa_value = 1.0 at the altitudes which the bulk of a fire's smoke plume is likely to be observed at after several hours
of burning (approximately 1500-2000 m). We therefore hypothesise that a restriction on qa_values would have minimal impact
on TCCO retrievals in our dataset, particularly as two higher resolution datasets were already used to remove cloud affected
plumes (the FRP-PIXEL Quality Product and VIIRS visible imagery; 3 km and 750 m at nadir respectively). Similarly to the
validation of the FMD procedure described above,  S5P CO qa_values were evaluated using the TCCON ground-based solar
FTIR monitoring networks, with this conducted by averaging the S5P CO product over 50 km$^2$ areas around each ground
measurement site and comparing the TCCO values Borsdorff et al. (2014). None of these sites were located in Africa, and the
validation was based on large area averaging of ambient-type total column CO data from clear-sky conditions (Borsdorff et
al., 2014). Our matchup dataset features TCCO observations over far smaller areas with extremely elevated CO and strong
spatial CO gradients. The application of the qa_values applied in Borsdorff et al. (2014) is therefore unlikely to be fully
representative of our application, and we hence treat the qa_values assigned in the S5P data with caution and choose not to
apply a qa_value threshold to the S5P data used to derive the $EC_{CO}^b$ values of the main work. We have, however, calculated a
set of  $EC_{CO}^b$ values from S5P data to which a qa_value > 0.5 threshold was applied and these are shown in Figure A2. These
values are summarised in Table A1 along with the $EC_{CO}^b$ values derived when the previously discussed Fixed Mask De-striping
(FMD) is applied, and the respective percentage difference between these and the $EC_{CO}^b$ values presented in the main text. The
qa_values > 0.5 thresholding results in an average $EC_{CO}^b$ value difference of only 2.4% across all five biomes (maximum
difference is 5.9%) demonstrating that the impact of the mid-level cloud on the retrieval of TCCO in our study is rather minimal
and well within the uncertainty bounds of the $EC_{CO}^b$ values.


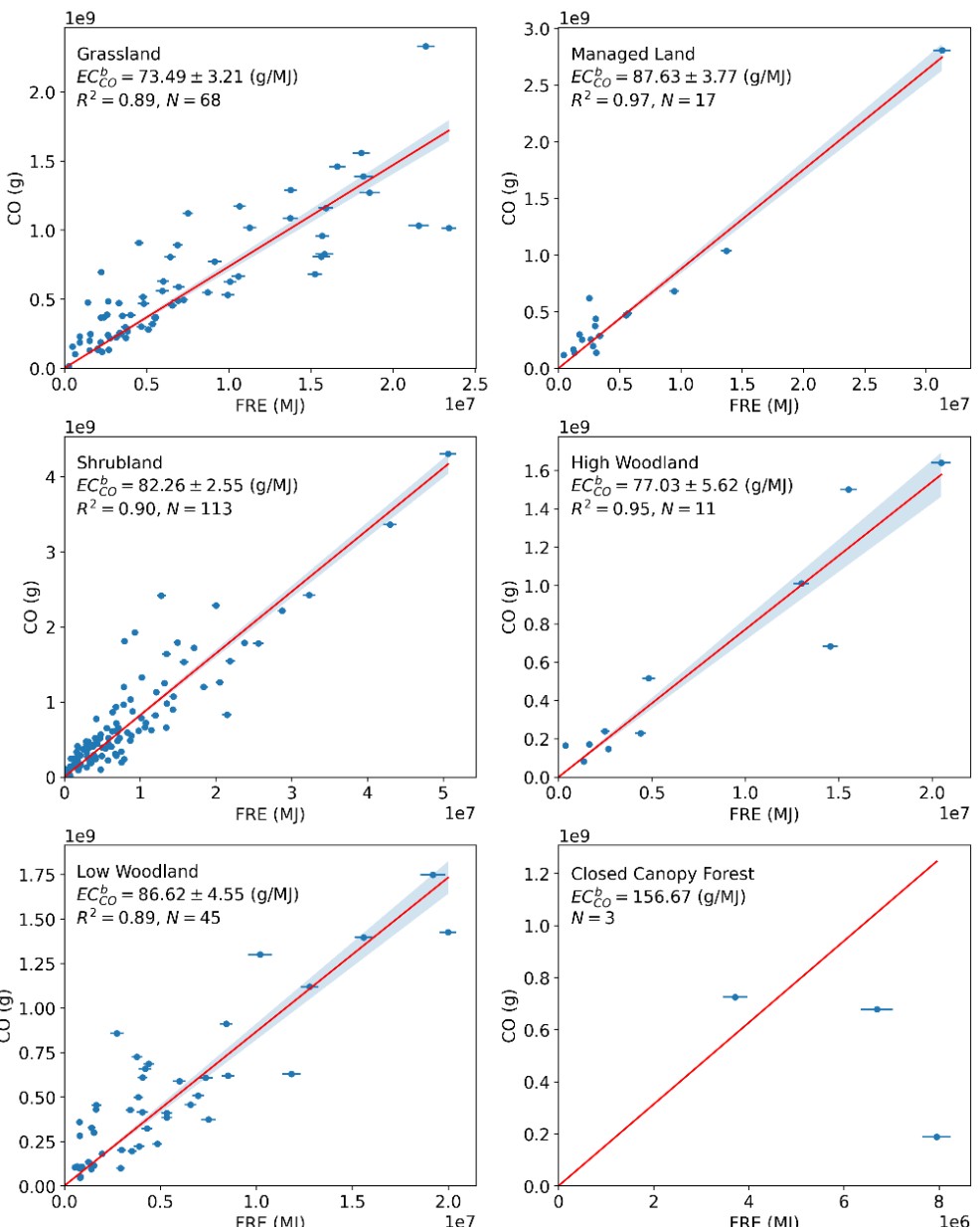

**Figure A2. CO Emissions coefficients $[EC_{CO}^b]$ derived with a qa_value > 0.5 threshold applied to the S5P TCCO data.**


**Table A1: Comparison of smoke emission coefficients generated from standard S5P data (which has neither the FMD procedure applied or a qa_value threshold of > 0.5 applied) and those generated from S5P with both these adjustments applied. Percentage differences between the standard S5P data and these two variants of the S5P data are also detailed.**

| | S5P without FMD applied & including all qa_values | S5P with FMD applied | Percentage difference (%) | S5P excluding qa_value < 0.5 | Percentage difference (%) |
|---|---|---|---|---|---|
| *Grassland* | 75.51 | 68.94 | -8.7 | 73.49 | 2.7 |
| *Shrubland* | 81.07 | 76.16 | -6.1 | 82.26 | -1.5 |
| *Managed land* | 88.35 | 82.91 | -6.2 | 87.63 | 0.8 |
| *High-woodland savanna* | 81.85 | 79.99 | -2.3 | 77.03 | 5.9 |
| *Low-woodland savanna* | 85.49 | 77.04 | -9.9 | 86.62 | -1.3 |


## 6.2 Appendix B

To maintain a consistent methodology between the FREM CO-based fire emissions inventory described in this work and the TPM-based version described in Nguyen and Wooster (2020) which was derived using Orthogonal Distance Regression (ODR), the OLS regression approach used herein was re-applied to the fire-plume match-up dataset of Nguyen and Wooster (2020). Updated FREM TPM-based fire emissions coefficients $[C_e^{TPM}]$ were generated from the nearly 1000 sample fires detailed in Nguyen and Wooster (2020). Each matchup consisted of a set of SEVIRI FRP-PIXEL product AF pixels for the target fire, along with the 1 km MCD19A1 MAIAC AOD product for that fire (see Nguyen and Wooster (2020) for details). Figure B1 shows the updated TPM emissions coefficients $[C_e^{TPM}]$ for each of the six biomes defined herein, and these are summarised in Table B1Table (Col 1) along with the previous ODR-derived values of Nguyen & Wooster (2020; Col 2) and various other forms of the same coefficients.

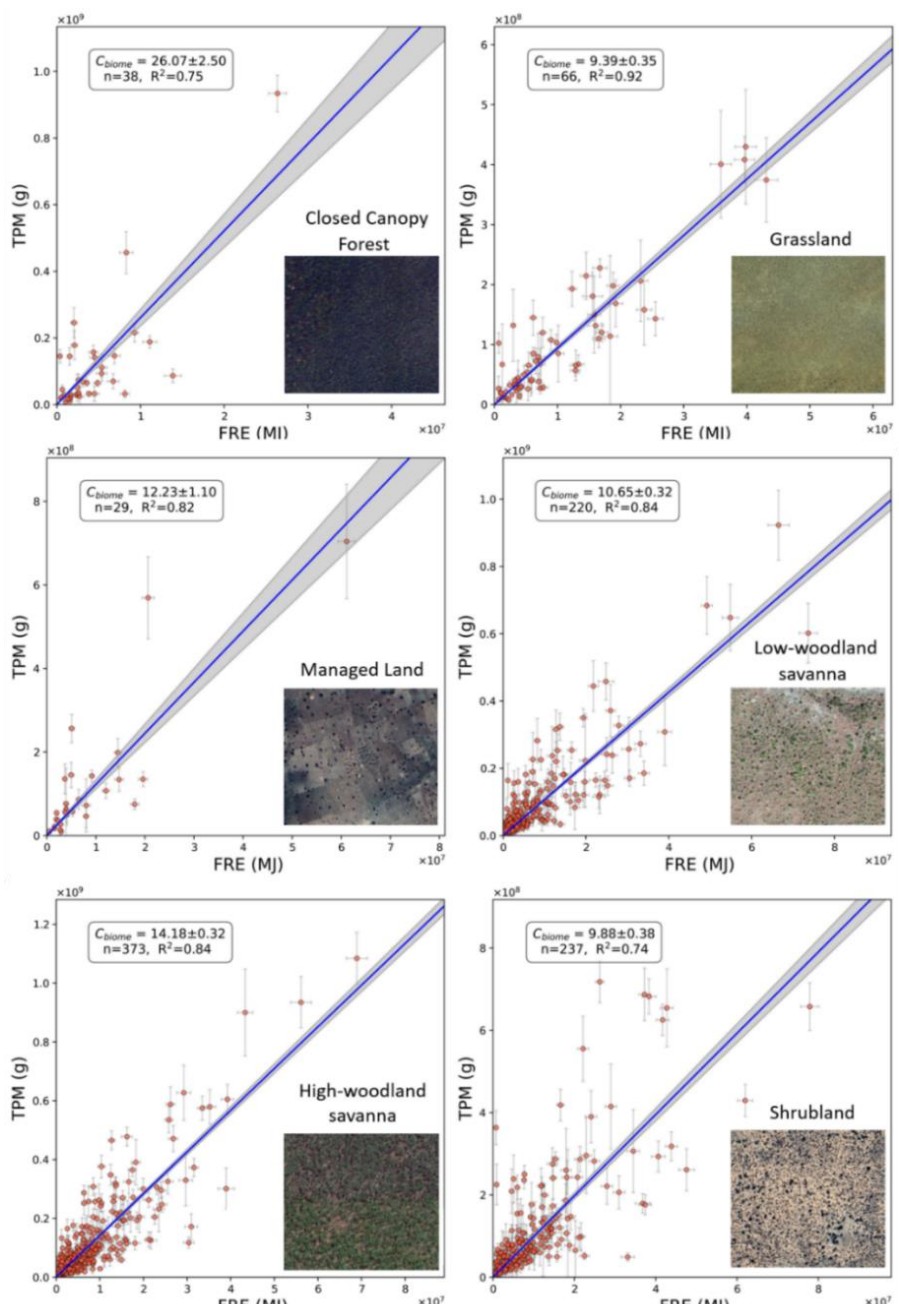

**Figure B1. TPM Smoke emission coefficients ($C_e^{TPM}$; in g.MJ-1) for the six African fire-affected biomes defined in the main manuscript, each derived from the slope of an ordinary least-squares (OLS) regression between data of fire-emitted total particulate matter (TPM) and matching fire radiative energy (FRE). The grey shaded area defines the 95% probability prediction interval of the OLS-derived slope. Each scatterplot is accompanied by an illustrative insert that depicts the typical landcover for the biome as seen in ©Google Earth (example locations are Closed Canopy Forest 10.359° S, 19.086° E; Grassland 21.180° S, 19.560° E; Managed Land 10.495° N, 7.586° E; Low-Woodland Savanna 7.085° N, 27.095° E, High-Woodland Savanna 12.523° S, 23.323° E and Shrubland 23.055° N, 22.242° E).**


**Table B1. TPM emission coefficients from previous FREM versions and updates (in units of g.MJ$^{-1}$)**

|  | FREM $C_e^{TPM}$ (OLS updated) | FREM $C_e^{TPM}$ (ODR; Nguyen and Wooster, 2020) | FREM $C_e^{TPM}$ (Mota & Wooster, 2018) | FREM $C_e^{TPM}$ (from $C_e^{CO}$ of Figure 2 and Equ 1) | FEER- Equivalent (see Nguyen and Wooster, 2020) |
|---|---|---|---|---|---|
| Closed canopy Forest | 26.07 | 34.33 | 65.63 | 16.43 | 16.34 |
| Managed land | 12.23 | 13.98 | 15.62 | 15.00 | 15.80 |
| Grassland | 9.39 | 9.99 | 13.03 | 9.52 | 10.98 |
| Shrubland | 9.88 | 12.17 | 17.36 | 10.22 | 10.97 |
| Low-woodland savanna | 10.65 | 12.10 | 19.75 | 10.78 | 12.78 |
| High-woodland savanna | 14.18 | 16.43 | 19.75 | 10.32 | 13.81 |


**Table B2. Results detailing the statistical significance of the $EC_{TPM}^b$ values of each biome pair. P-values below 0.05 are coloured green, those below 0.15 are coloured yellow, while any biome pair with p-values about 0.15 are coloured red.**

| Biome 1 | Biome 2 | p-value | t-value |
|---|---|---|---|
| grassland | managed land | 0.02 | 2.46 |
| grassland | shrubland | 0.34 | 0.95 |
| grassland | high woodland savanna | 0.00 | 10.10 |
| grassland | low woodland savanna | 0.01 | 2.66 |
| grassland | closed canopy forest | 0.00 | 6.61 |
| managed land | shrubland | 0.04 | 2.02 |
| managed land | high woodland savanna | 0.09 | 1.70 |
| managed land | low woodland savanna | 0.17 | 1.38 |
| managed land | closed canopy forest | 0.00 | 5.06 |
| shrubland | high woodland savanna | 0.00 | 8.66 |
| shrubland | low woodland savanna | 0.12 | 1.54 |
| shrubland | closed canopy forest | 0.04 | 2.01 |
| high woodland savanna | low woodland savanna | 0.00 | 7.80 |
| high woodland savanna | closed canopy forest | 0.00 | 8.65 |
| low woodland savanna | closed canopy forest | 0.00 | 6.11 |


## 6.3 Appendix C

**Table C1. Summary of WRF-CMAQ model configuration**

| General features | |
| --- | --- |
| Domain extent | 10°E - 44 °E, 5°S -32°S |
| Modelled time period | 15th June to 28th July, and 29st July to 29st Aug 2019 |
| Resolution | 9 km × 9 km, 35 vertical levels (top layer at 5 kPa) |
| **WRF configuration** | **Scheme** |
| cloud microphysics | Lin et al. |
| radiation (shortwave) | Goddard |
| radiation (longwave) | Rapid Radiative Transfer Model (RRTM) |
| boundary layer physics | Mellor-Yamada-Janic (MYJ) |
| land surface processes | Noah LSM |
| cumulus convection | Grell 3-D |
| **CMAQ configuration** | |
| Chemistry mechanism | CB6r3 |
| aerosol module | AERO7 |
| Dust emissions | inline |
| Biogenic emissions | inline BEIS3 |
| **Initial and boundary conditions** | |
| Metrology | NCEP FNL, 0.25° × 0.25°, 26 levels, 6 hour |
| Chemistry | WACCM, 0.9° × 1.25°, 88 levels, 3 hour |

## 6.4 Appendix D

The updated TPM emissions coefficients $[C_e^{TPM}]$ calculated in Appendix B (with the exception of the closed canopy forest value) were used to derive an emission inventory for aerosols that was then used as an input to a WRF-CMAQ simulation. These same simulations used gaseous emissions generated from the FREMs_bCO emissions coefficients described in the main article as input. To evaluate the TPM emissions values, the AOD fields produced by these CMAQ simulations were compared with independent ground based and satellite-based AOD metrics. The WRF-CMAQ model set-up and configuration are described in Appendix C and in the main article, while the results of the FREM-TPM emissions estimates evaluation are presented in here.

AERONET is a global network of ground-based sun photometers that provides retrievals of aerosol optical properties, including Angström exponent, aerosol refractive index, and aerosol optical depth (AOD) at different wavelengths (Holben et al., 2001). Data from AERONET sites within the CMAQ model domain were used in comparisons with CMAQ-generated AOD fields (at 381nm). The AERONET sites used were: Maun Tower (19.9°S, 23.55°E), Lubango (15.0°S, 13.4°E), Misamfu (10.2°S, 31.2°E), Gobabeb (23.6°S, 15.0°E), Welgegund (26.6°S, 26.9°E) and Skukuza (35.0°S, 31.6°E). AERONET AOD data is available for the full simulation period from each of these sites, with the exception of Misamfu and Welgegund that have data available from 15$^{th}$ June until 29$^{th}$ July and 13$^{th}$ August respectively. AERONET AOD observations at 380 nm are used in comparisons to WRF-CMAQ modelled AOD at 381 nm.

Figure D1 shows hourly mean AOD, averaged across all six AERONET sites, as determined by CMAQ and by the AERONET measurements. CMAQ AOD captures the temporal pattern of AOD rather well across these six sites, but in general tends to show higher values than the ground-based measures. Hourly modelled and observed AOD were compared in terms of their Pearson's correlation (r) and NMBF in each month as well as over the full simulation period, and these results are summarised in Table D1 along with monthly mean AOD for CMAQ and AERONET at each site. NMBF over the full simulation period at the six AEONET sites ranges between a 4% underestimation and 41% overestimation by CMAQ relative to AERONET, and the Pearson's correlation coefficient ranges between 0.36 and 0.72.

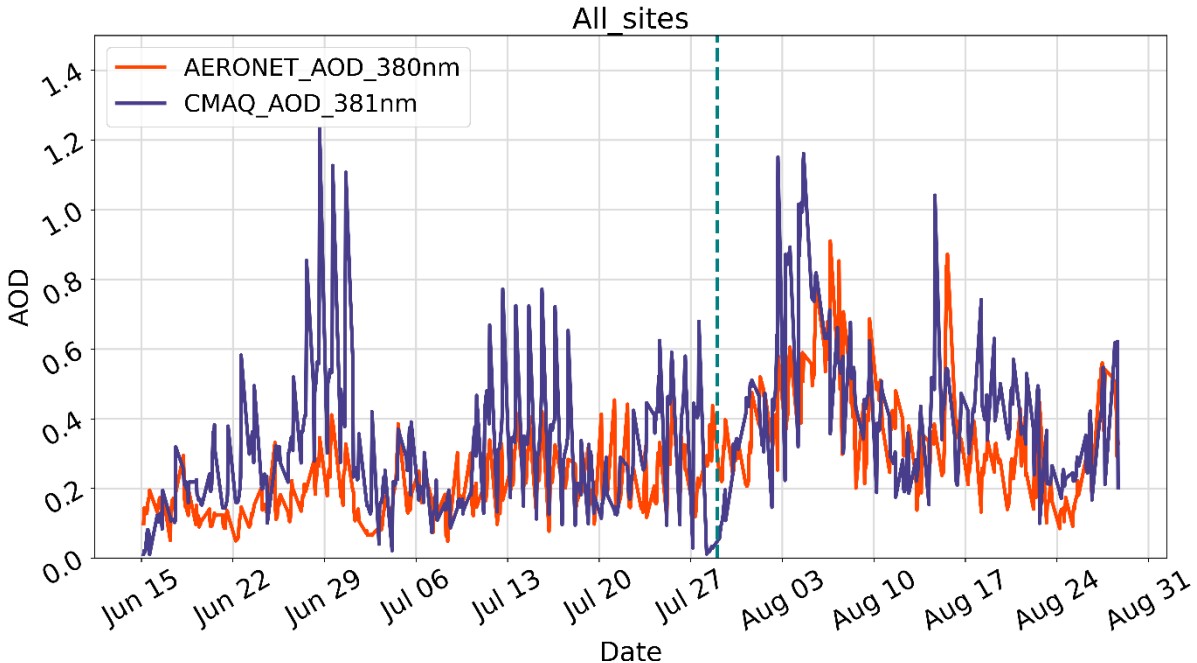

**Figure D1. Hourly AOD averaged over all six AERONET sites, both from the CMAQ simulations and AERONET observations. A vertical dotted line on 29$^{th}$ July indicates the start of the seconds simulation.**

**Table D1. Monthly mean of hourly CMAQ and AERONET AOD, the NMBF of hourly CMAQ AOD with respect to observations and the temporal Pearson's correlation coefficient of hourly AOD over each month and the whole modelled period (15th June to 29th August 2019).**

| | Month | CMAQ Mean | AERONET Mean | NMBF | Pearson's Correlation (r) |
|---|---|---|---|---|---|
| Lubango | June | 0.52 | 0.26 | 1.00 | 0.71 |
| | July | 0.68 | 0.49 | 0.65 | 0.70 |
| | August | 0.66 | 0.56 | 0.16 | 0.42 |
| | All | 0.64 | 0.45 | 0.41 | 0.50 |
| Misamfu | June | 0.41 | 0.13 | 2.08 | 0.72 |
| | July | 0.20 | 0.22 | -0.70 | 0.16 |
| | All | 0.27 | 0.19 | 0.47 | 0.13 |
| Gobabeb | June | 0.22 | 0.17 | 0.34 | 0.75 |
| | July | 0.20 | 0.25 | -0.30 | 0.60 |
| | August | 0.21 | 0.24 | -0.11 | 0.13 |
| | All | 0.21 | 0.22 | -0.03 | 0.45 |
| Maun Tower | June | 0.06 | 0.10 | -0.55 | 0.57 |
| | July | 0.12 | 0.12 | -0.04 | 0.69 |
| | August | 0.40 | 0.39 | 0.05 | 0.61 |
| | All | 0.21 | 0.22 | 0.00 | 0.72 |
| Welgegund | June | 0.13 | 0.12 | 0.10 | 0.57 |
| | July | 0.13 | 0.11 | 0.31 | 0.50 |
| | August | 0.52 | 0.45 | 0.14 | 0.29 |
| | All | 0.21 | 0.18 | 0.21 | 0.66 |
| Skukuza | June | 0.27 | 0.21 | 0.26 | 0.53 |
| | July | 0.36 | 0.25 | 0.43 | 0.19 |
| | August | 0.43 | 0.40 | 0.08 | 0.41 |
| | All | 0.37 | 0.29 | 0.27 | 0.39 |
| All Sites | June | 0.28 | 0.16 | 0.78 | 0.64 |
| | July | 0.28 | 0.22 | 0.27 | 0.33 |
| | August | 0.44 | 0.37 | 0.20 | 0.53 |
| | All | 0.34 | 0.26 | 0.31 | 0.59 |

In addition to this comparison to ground-based AOD data, CMAQ modelled AOD at 550 nm was compared to the MODIS MAIAC 550 nm 1 km product (Collection 6 MCD19A2; Lyapustin et al., 2018) - the same AOD product used in the derivation of FREMv2 TPM emissions coefficients (Appendix B and Nguyen and Wooster (2020)), though a completely different set of days were used in the generation of the matchup dataset. Daytime Aqua and Terra overpasses occurring between approximately 08:00 and 10:00 UTC daily over the CMAQ domain were compared to mean CMAQ AOD between 08:00 and 10:00 at 550 nm. Both modelled and observed AOD were remapped to a 0.1°×0.1° grid for ease of comparison.

The spatial distribution of monthly mean AOD in the MAIAC AOD product and CMAQ is shown in Figure C2. Most notable in Figure C2Figure is the large variation between under and over estimation by modelled AOD compared with MAIAC AOD, as can be seen in the difference plot of C2c. In the north west of the domain, where the highest fire activity occurs (See main article, Figure 8 c), some areas feature CMAQ AOD that is close to 60% greater than MAIAC AOD, with the highest overestimation occurring in June. While in other regions of the of the domain, CMAQ underestimates observed AOD significantly. In these areas, however, AOD values are already low and hence, this supposed underestimation is not as significant in absolute terms, though it does indicate that - in its base state - the CMAQ model tends to underestimate AOD.

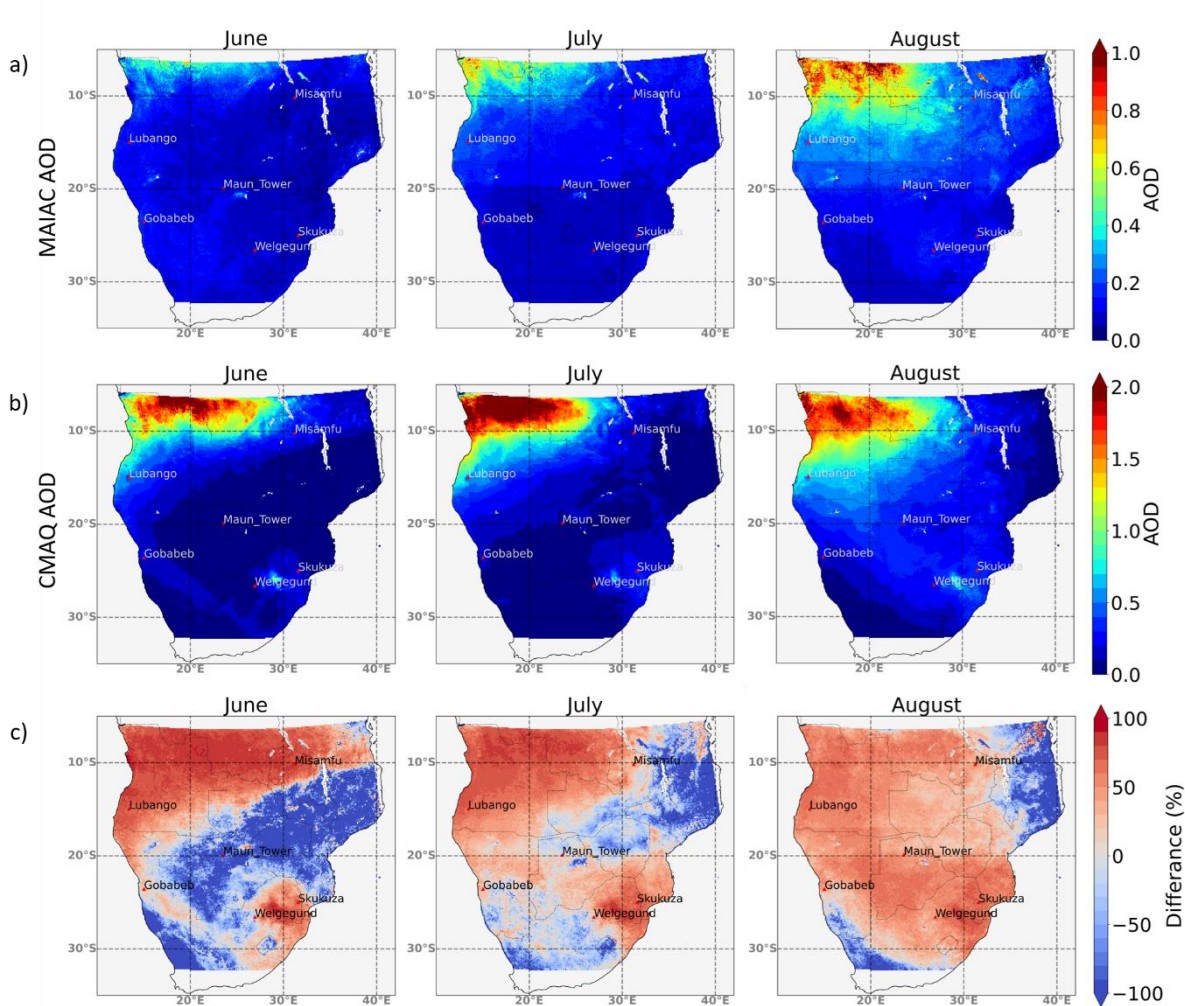

**Figure D2. Mapped mean monthly AOD at 550 nm from a) the MAIAC satellite product and b) CMAQ simulations (note - colour scale differences between a) and b)) during the simulation period from 15th June to 29th August 2019. c) shows the percentage difference between MAIAC and CMAQ AOD**

Daily modelled and observed AOD in each ROI (see main article Figure 8a) and in the full domain were used to generate mean monthly AOD during the simulation period (Table C2) and the NMBF and Pearson's correlation coefficient (r) between CMAQ and MAIAC daily AOD were also calculated. The results show that CMAQ AOD, in general, is significantly overestimated relative to MAIAC AOD, and this overestimation is greater than for the CMAQ TCCO comparisons to Sentinel-5P TCCO shown in the main manuscript (4). Daily mean CMAQ AOD in the domain for the full simulation period is 120% higher than MAIAC mean AOD, and when restricted to days in June this increases to 184%. Mean CMAQ AOD in ROI1 – which includes much of the area with the highest fire activity - shows the largest overestimation, ranging between 105% and

735 180% depending on the month. Conversely ROI2 and ROI3, in which there is generally lower fire activity, show lower NMBF values ranging from an underestimation of 30% to an overestimation of 77% by CMAQ AOD. The correlation between modelled and observed daily means varies by ROI and by month, but in most cases r > 0.60.

**Table D2. Monthly means of daily CMAQ and MAIAC AOD, in the full extent of the domain and the ROIs, the NMBF of daily**
**CMAQ AOD with respect to observations and the temporal Pearson's correlation coefficient of daily AOD over each month and the whole modelled period are also included (15th June to 29th August 2019)**

| | | CMAQ Mean AOD | MAIAC Mean AOD | NMBF | Pearson's Correlation (r) |
|---|---|---|---|---|---|
| Full Domain | June | 0.11 | 0.04 | 1.84 | 0.77 |
| | July | 0.13 | 0.05 | 1.37 | 0.63 |
| | August | 0.15 | 0.79 | 0.94 | 0.21 |
| | All | 0.13 | 0.06 | 1.2 | 0.5 |
| ROI1 | June | 0.26 | 0.09 | 1.8 | 0.9 |
| | July | 0.37 | 0.15 | 1.42 | 0.73 |
| | August | 0.55 | 0.27 | 1.05 | 0.48 |
| | All | 0.41 | 0.18 | 1.21 | 0.64 |
| ROI2 | June | 0.07 | 0.09 | -0.3 | 0.44 |
| | July | 0.07 | 0.07 | 0.06 | 0.69 |
| | August | 0.21 | 0.11 | 1 | 0.63 |
| | All | 0.13 | 0.09 | 0.42 | 0.64 |
| ROI3 | June | 0.07 | 0.04 | 0.77 | 0.8 |
| | July | 0.11 | 0.09 | 0.16 | 0.79 |
| | August | 0.22 | 0.16 | 0.39 | 0.59 |
| | All | 0.14 | 0.1 | 0.32 | 0.73 |
| ROI4 | June | 0.07 | 0.05 | 0.56 | 0.69 |
| | July | 0.11 | 0.05 | 0.97 | 0.46 |
| | August | 0.17 | 0.07 | 1.41 | 0.71 |
| | All | 0.12 | 0.06 | 1.13 | 0.81 |

As with the evaluation conducted for the FREM-derived CO emissions (see main article Section 4), comparisons between CMAQ and MAAIC AOD were also conducted for individual smoke plumes identifiable in the MAIAC AOD product. Individual fire emitted plumes were identified in the MAIAC AOD product at its native 1 km spatial resolution. Polygons were used to define plume boundaries, and each plume was matched between modelled and observed AOD data. A 0.1° grid cell buffer was applied to account for variations in the spatial distributions of the plumes. Fire emitted AOD for CMAQ and MAIAC plumes were calculated via the method described in Nguyen & Wooster (2020).

Figure C3 shows the spatial distribution of the 415 individual smoke plumes used in comparisons and the relation between the fire emitted AOD fields of CMAQ and MAIAC for each of these plumes. There is a large spread in the data, and the Pearson's correlation is relatively low at 0.43. The NMBF indicates an overall underestimation of CMAQ AOD compared with MAIAC AOD, by 10%. This is drastically different from the 120% overestimation of daily AOD by CMAQ relative to MAIAC in the full domain comparisons. The impact of model error, and the more extensive spatial variability between the modelled and observed plumes, may significantly contribute to the large differences seen in the comparison made at large scales and those for individual plumes. The true accuracy of the FREM derived emission is likely somewhere in between.

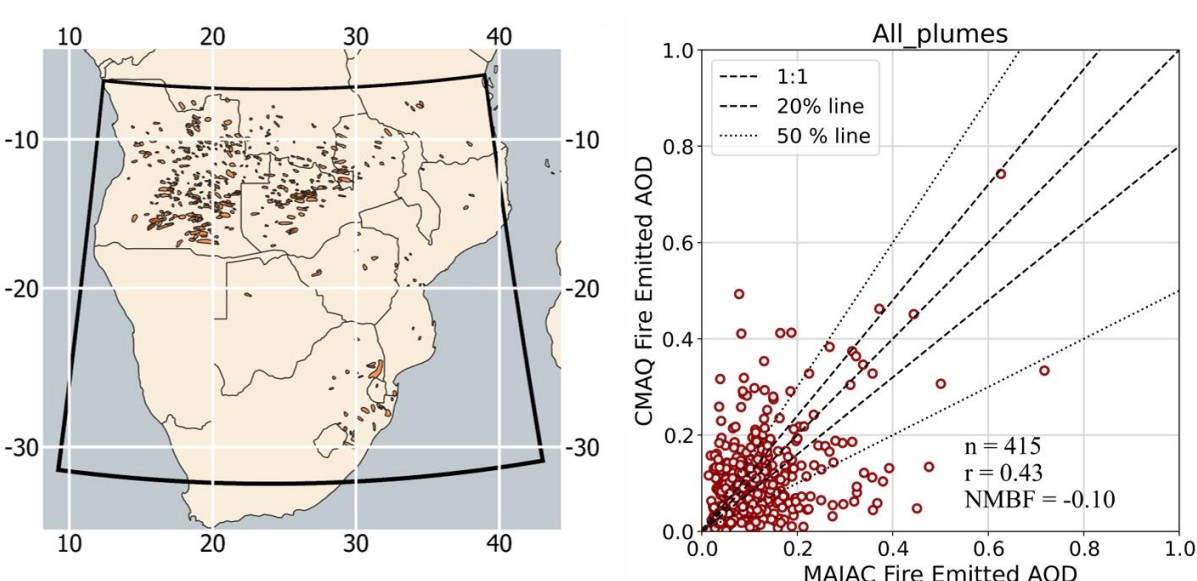

**Figure D3. Location of smoke plumes identified in the MAIAC 1 km AOD product between 15th June and 29th August 2019 and the bounding polygons /used to define the area over these plumes (left). Relationship between fire-emitted CMAQ AOD and observed fire-emitted MAIAC AOD in these plumes (right), the Pearson's correlation and NMBF of the dataset is shown along with dotted lines indicating the 1:1, 20% and 50% lines.**

# 7  Code availability

Code is available upon request to Hannah Nguyen (hannah.nguyen@kcl.ac.uk)

# 8  Data Availability

The fire emissions inventory presented will be available from the EUMETSAT Land Surface Analysis Satellite Application Facility (http://landsaf.meteo.pt) in the near future. Data can be provided upon request to Hannah Nguyen (hannah.nguyen@kcl.ac.uk) in the interim.

# 9  Author contributions

775 MW was responsible for conceptualization of methodology, and JP was responsible for the generation of the CO plume match-up dataset and emission coefficients. HN assisted on emissions methodology and was responsible for emission validation with WRF-CMAQ. MW and HN contributed to first draft and prepared initial visualisations. HN and MW were responsible for writing and editing the final manuscript.

# 10  Competing Interests

The contact author has declared that neither they nor their co-authors have any competing interests.

# 11  Acknowledgements

Sentinel-5P products are distributed freely by Copernicus Open Access Hub (https://scihub.copernicus.eu/), as is the SEVIRI FRP-PIXEL product of the EUMETSAT LAS SAF (https://landsaf.ipma.pt/) and VIIRS product of LAADS DAAC (https://ladsweb.modaps.eosdis.nasa.gov/). We would like to thank the LAS SAF and MODIS teams for the development and
785 distribution of these products, as well as the developers and providers of FEER, GFAS, GFED, CCI Landcover, Landsat VCF and FireCCISFD11. Funding for this work was provided by EPSRC's Centre for Doctoral Training in Cross-Disciplinary Approaches to Non-Equilibrium Systems (CANES; EP/L015854/1), NERC National Capability funding to NCEO (NE/R016518/1), NERC Grant (NE/S014004/1) and by EUMETSAT's Satellite Applications Facility Programme which supports the Land Surface Analysis Satellite Application Facility (https://landsaf.ipma.pt/en/, LSA SAF CDOP-4) where the
790 Meteosat FRP-PIXEL and FRP-GRID products are generated. We also thank the anonymous reviewers for their comments which have helped improve the quality and content of this manuscript.

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
