# Peer review of "Biomass burning CO, PM and fuel consumption per unit burned area estimates derived across Africa using geostationary SEVIRI Fire Radiative Power and Sentinel-5P CO data"

_Atmospheric Chemistry and Physics, 2022_

## Author Comment (AC1)

**CCs - Patrick Freeborn**

We thank Patrick for the time taken and his valuable comments. We have taken these onboard and incorporated the suggested changes into the manuscript. We feel that this has greatly improved the clarity and substance of the work. Green highlighted comments have been addressed in the revised manuscript.

Specific Comments:

1. With some effort I was able to understand the methods and results, but not with any help from the nomenclature. In my view, the nomenclature used herein is internally inconsistent and may conflict with the nomenclature used in previous works describing older versions of the Fire Radiative Energy Emissions (FREM) approach. Please note the following:

- a. The "smoke emission coefficient" is introduced on Page 2, Line 63, as $C^x_e$, where the superscript $x$ indicates the trace gas or aerosol of interest. The subscript e is never defined, though I suspect the subscript e stands for "emission" and the $C$ stands for "coefficient". However on Page 3, Line 72, the subscript e in $\beta_e$ stands for extinction. Nevertheless, the emissions coefficients for total particulate matter (TPM) and carbon monoxide (CO) are appropriately written as $C_e^{TPM}$ and $C_e^{CO}$.

- b. The emissions factor ($EF$) is introduced in Equation 1 on Page 3, Line 86. However for emissions factors, the subscript $x$, not the superscript $x$, indicates the trace gas or aerosol of interest. To better align with the nomenclature of the smoke emission coefficient, it seems to me that $x$ belongs as a superscript in the nomenclature of the emissions factors. Also, if $EF$ stands for emissions factor, then it seems logical to me that $EC$ should stand for emissions coefficient.

- c. On Page 3, Line 87, it states: "Where is the biome-specific emission coefficient for trace gas species $x$". Okay, so the subscript $e$ doesn't stand for "emission" but rather $e$ indicates the biome? Why not use a subscript $b$ instead of an $e$ to indicate the biome?

- d. Also on Page 3, Line 87, it states: "$EF_x$ is the species $x$ emission factor for that biome". However there is no subscript $e$ or $b$ in $EF_x$ that indicates the biome?

- e. On Page 5, Lines 133-135, is states: "the FREM methodology derives a biome-dependent 'smoke emission coefficient' for a reference species $y$, [$C_e^y$] from the relationship between the thermal energy a fire radiates (i.e. the FRE in MJ) and the mass of the target compound $y$ it emits (in kg or g)". What is the difference between the reference species $reference$, the reference species $y$, and the target compound $y$? To the point, what is the

difference between $C_e^{reference}$ on Page 3, Line 88, and $C_e^y$ on Page 5, Line 134? What is the difference between a reference species and a target compound?

- f. On Page 8, Figure 2, all of the legends in the subplots contain $C_{biome}$ = regression fit. This seems like new nomenclature, but not new if *e* = *biome*.

- g. Page 7, Lines 196-198: And here "*b*" is introduced as the base or reference? What is the difference between base and reference?

- h. Ultimately, it seems to me that Equation 1 could be written as follows:

$$EC_b^x \; [g.MJ^{-1}] = \frac{EF_b^x [g.kg^{-1}]}{EF_b^{ref} [g.kg^{-1}]} EC_b^{ref} \; [g.MJ^{-1}]$$

where *EC* is an emission coefficient, *EF* is an emission factor, the subscript *b* is the biome, the superscript *x* represents the trace gas or aerosol of interest, and the superscript ref represents the reference species.

We greatly appreciate the comments regarding nomenclature, and on reflection agree this could be clearer – although we tried to keep with "normal" convention.

The use of $C_e$ to denote smoke emissions coefficient was used in the submitted manuscript to align with the notation adopted in the first paper to show a top-down FREM-like emissions methodology. This was that which details the "FEER" fire emissions inventory (Ichoku & Ellison, 2014), and the same nomenclature was used by us in the papers describing the FREM technique (Mota & Wooster, 2018; Nguyen & Wooster, 2020). However, we do recognise that the discussion of how other species emissions are generated via Equation 1, and references to emissions factors (which are typically written $EF_x$, where x is the species of interest), has perhaps made this particular notation more confusing than helpful.

We have taken onboard the comments and redefined our notation as suggested by the reviewer. We adopt the notation $EC_X^b$, where $b$ is the biome and $x$ is the reference species. The smoke mass extinction coefficient remains $\beta_e$ – and this is how is it most commonly referred to in the aerosol science literature.

2. Page 5, Section 2.2: The calculation of total plume CO [in g or kg] is not altogether clear to me and could be better described in the methods. I think the units stored in the Sentinel-5 precursor/TROPOMI Level 2 CO Product are mol m$^{-2}$, and so I'm assuming a conversion using molar mass is performed, but this is never stated. What is the buffer size? It seems to me that the plume boundaries were hand digitized, so is it possible that the buffer sizes are not constant but change from matchup to matchup? I'm assuming the minimum CO value was simply subtracted from every pixel within the plume buffer to yield the excess CO above the

background, but this is never explicitly stated? I don't think "summing this excess over all plume pixels thus provided the total amount of fire-emitted CO in the plume for that matchup fire" is an accurate description. At some point you need to multiply the pixel value (mol m$^{-2}$ or g m$^{-2}$) by the pixel size (m$^2$), correct? What is the actual pixel size? Is the pixel size constant or does it vary depending on the location of the pixel in the swath? I believe earlier in the archive it's 7x7 km, correct? I also think the instrument settings for TROPOMI were changed on August 6$^{th}$, 2019, which reduced the along-track pixel dimensions to 5.5x7.0km. Were the different pixel sizes between 2018 and 2020 taken into account?

Thank you for these valuable comments. The method used in our manuscript in relation to the calculation of total emitted CO is very similar to that used in the calculation of fire emitted total particulate matter (TPM) in our previous papers (Mota & Wooster, 2018; Nguyen & Wooster, 2020). As such, a more detailed description was not included initially – but we recognise now that inclusion of a more explicit description of how plume total CO is calculated would be valuable – so the below has now been added -- Section 2.2 (line 146):

[revised manuscript text omitted]

3. Page 6, Section 2.3: Why the switch from ODR to OLS? Seems like 14% is a substantial difference just based on the selection of the regression method?

Thank you for this comment, a more in-depth discussion of this methodological choice has been added to section 2.3 (line 198):

"Matchup data for each 'fire biome' are shown in Error! Reference source not found. and were used to derive the set of biome-dependent CO smoke emission coefficients [$EC_{CO}^b$] listed in **Error! Reference source not found.**. Zero-intercept ordinary least squares (OLS) regression was used for this, rather than the orthogonal distance regression (ODR) used by Nguyen and Wooster (2020) during derivation of TPM smoke emission coefficients [$EC_{TPM}^b$]. OLS was used in this work for two main reasons. Firstly, although the ODR method considers the uncertainty in each of the variables, these uncertainties are themselves rather poorly constrained, with the known uncertainties only representing part of the total uncertainty sources. There are contributions to the uncertainty of FRP that are not quantifiable, for example due to variations in the amount of interception of a fire's FRP signal by any overlying tree canopy. We therefore deemed use of a regression method in which the slope is strongly driven by datapoint uncertainty to be unsuitable for use. Secondly, weighting based on uncertainty often resulted in undue weight being given to high value points (e.g. match-ups with high FRE and high plume-species amounts) due to them typically having lower relative uncertainties

(see Wooster et al., 2015). Due to their typically being very few high value datapoints in each 'fire biome' due to the heavy tailed nature of fire size distribution (Freeborn et al., 2009), these few large fires were potentially being too strongly weighted in the resulting calculation of $EC_x^b$. For these reasons, we opted to use OLS regression, and to ensure a consistent methodology for emission coefficient derivation we also applied the same approach to the Nguyen and Wooster (2020) dataset to re-derive their $EC_{TPM}^b$ values using OLS regression (see **Error! Reference source not found.**). The updated $EC_{TPM}^b$ for closed canopy forest, managed land, grassland, shrubland, low-woodland savanna and high-woodland savanna are 26.07 g.MJ$^{-1}$, 12.23 g.MJ$^{-1}$, 9.39 g.MJ$^{-1}$, 9.88 g.MJ$^{-1}$, 10.65 g.MJ$^{-1}$, and 14.18 g.MJ$^{-1}$ respectively. On average these new values are 14% lower than those reported in Nguyen and Wooster (2020) derived via ODR, and are the ones referred to and used hereafter. The WRF-CMAQ model-based approach to evaluating our final CO emissions rates and totals in Section **Error! Reference source not found.** was also used to carry out an analogous evaluation of the TPM emissions generated from the updated $EC_{TPM}^b$ values of **Error! Reference source not found.** (see **Error! Reference source not found.**)."

4. Page 7, Lines 186-194: What are the consequences of this limitation? How much CO is emitted from closed canopy forests relative to the total CO budget for Africa? I could not find contributions from individual biomes reported anywhere. Would it even be possible to complete the "FREM_bCO" emissions inventory without leveraging the "FEER-equivalent" value for closed canopy forests? Will this limitation appear again in other closed canopy forests across the globe when matching TROPOMI TCCO retrievals with different geostationary AF and FRP product.

We thank the reviewer for this valuable comment, we have added a section of text discussing the cause of this smaller sample size in closed canopy forest along with a more detailed discussion demonstrating that this limitation is observed in all top-down emission inventories to date. We have also referenced the FRE contribution of closed canopy forest compared to other fire biomes, which was calculated from 6 years' worth of SEVIRI FRP data in Nguyen and Wooster (2020).

Using only the current CO match-up dataset generated in this work, delivery of a FREMs_bCO inventory without use of any $EC_{TPM}^b$ value for the closed canopy forest biome would not be possible. However, it is not essential to use a FEER-equivalent $EC_{TPM}^b$. The $EC_{TPM}^b$ for closed canopy forest shown in Appendix A could be used instead to generate TPM emissions – and then CO values derived from this using Equation 1. We selected to use a FEER-equivalent $EC_{TPM}^b$ as opposed to this latter approach for several reasons. Firstly, since FEER was generated from 7 years' worth of MODIS FRP data (compared to the 1 year of SEVIRI FRP data used for FREM $EC_{TPM}^b$) it includes a greater number of individual plumes from which an emission coefficient can be derived (Ichoku and Ellison, 2014). Additionally, the updated FREM $EC_{TPM}^b$ values for all other fire biomes of Appendix A were all within 30% of their FEER-equivalent $EC_{TPM}^b$ values, with the exception of closed canopy forest, which was 38% higher. It is also likely that FRP measurements from polar-orbiting sensors such as MODIS may be better suited to detecting fires in closed canopy forest due to their higher spatial

resolution and varying sensor view angles available at the same location. Due to tree canopy interception of surface emitted FRP preventing the full FRP signal from reaching the sensor (Roberts et al., 2018), the detection of fires in closed canopy forest is optimal at angles closer to nadir. Therefore, FRP products with a lower minimum FRP detection threshold and which sample a location at varying view angles (unlike geostationary satellites) potentially are able to capture a more representative FRP measure from fires in closed canopy forest.

An increased number fire match-ups in the closed canopy forest biome, and ideally in general in other biomes as well, is indeed necessary. At the time of this research only 2 years' worth of S5P CO observations were available, limiting the plume match-up dataset further, even beyond the limitations set by the labour-intensive manual selection methods used. Some improvements to the methods are being invested in to speed plume detection in future work. Machine learning techniques could be used for plume identification and digitisation to produce a more automated method that could be enacted far quicker on large amounts of data than the time intensive manual methods used herein. This could also potentially make the detection of different plumes more consistent and allow the overall FREM method to be efficiently extended to other geographic regions, whilst minimising the number of 'low sample size' fire biome matchups.

Additional paragraph has been added at Line 218:

"For each of the six fire biomes at least 12 matchup fires were identified for derivation of $EC_{CO}^b$, apart from for closed canopy forest. Tropical evergreen forests (the primary type of closed canopy forests in tropical regions) are generally not very susceptible to fire, except during periods of extreme drought, due to the high humidity and low windspeed within the dense forest canopy and the limited amount of surface fuel available due to rapid decomposition of surface litter in these environments (Marengo et al., 2011; Tomasella et al., 2013). Fires in such tropical forests are most often the result of human land-clearing activity and are typically small in size, unless heavy machinery is involved in land clearing (Van Leeuwen et al., 2014). Furthermore, FRP observations in closed canopy forest can be affected by tree canopy interception of surface emitted FRP (Roberts et al., 2018). These factors result in a lower number of observable and identifiable fire match-ups for tropical closed canopy forest areas. Smaller fire sizes and fewer match-up fires being acquired in closed canopy forests areas relative to other biomes was also observed during previous applications of the FREM approach (Mota & Wooster, 2018; Nguyen & Wooster 2020) and the FEER approach (Ichoku & Ellison, 2014), even when using 7 years' worth of MODIS FRP and AOD data in the latter case. In this work the ability to identify small fires in closed canopy forest is further limited by i) the spatial resolution of the S5P TCCO observations which are at least 5 times lower resolution than the 1 km AOD product used in Nguyen & Wooster (2020) and ii) the limited availability of the S5P trace gas products which only became operational form mid-2018. An increased timeseries of S5P data and the exploitation of machine learning methods such as object recognition may aid in identifying a greater number of plumes in closed canopy forest - and this the subject of ongoing work.

Due to the FEER emission inventory exploiting a far larger dataset from which to identify fire match-ups (7 years of MODIS FRP and AOD) and it obtaining many more fire match-ups in tropical closed canopy forest (Ichoku and Ellison, 2014) we instead derive $EC_{CO}^b$ for closed canopy forest from the 'FEER-equivalent' value. The method used to derive this is detailed in Nguyen and Wooster (2020), and essentially involves aggregating the FEER $C_e^{TPM}$ emission coefficients of Ichoku and Ellison (2014) (https://feer.gsfc.nasa.gov/data/emissions/) to the relevant fire biome. Equation 1 was then applied to obtain a FEER-equivalent $EC_{CO}^b$ , which was calculated as 156.7 g.MJ$^{-1}$ for the closed canopy forest fire biome. We generated FEER-equivalent $EC_{CO}^b$ for each of the other five fire biomes to compare these to our directly derived $EC_{CO}^b$ values, and found agreement within ±34% (see **Error! Reference source not found.**), somewhat justifying our use of the FEER-equivalent value in the closed canopy forest biome where a directly derived $EC_{CO}^b$ value was not achieved. Further, Nguyen and Wooster(2020) showed that mean monthly FRE contribution from fires in closed canopy forests does not exceed 10% of the total monthly FRE coming from African fires, and thus its total is of relatively low importance to continental-scale CO emissions totals."

5. Page 16, Lines 344-346: This sentence states: "Aerosol species emissions were generated through an analogues application of the updated FREM-TPM emission coefficients of Nguyen & Wooster (2020) (see Appendix A)". Given the title of this manuscript, I fail to see where this part fits into the FREM-Derived CO Emissions evaluation/assessment. Since this is not really related to the CO emissions inventory, and since all of the AOD results are compared in the Appendix and not in the body of the manuscript, I would strongly reconsider whether the TPM/AOD evaluation/assessment is necessary.

It is true that the results of Appendix A are not directly used in the main text, but we believe these updates are important and highly relevant to this manuscript as they use the same underlying FREM methodology. We have however adjusted the title and abstract of the manuscript to be more reflective of the content as a whole. Thank you for this suggestion.

New title:

**"Biomass burning CO, PM and fuel consumption per unit burned area estimates derived across Africa using SEVIRI Fire Radiative Power (FRP), Sentinel-5P CO and MODIS AOD observations"**

6 Page 19, Line 382: Results of the evaluation/assessment from this point forward are extremely confusing to me. I think part of the reason is that the authors use the term "daily summed TCCO". (Figure 10) From my understanding there is one map of modelled TCCO and one map of S5P- observed TCCO generated daily sometime between 12:00 and 14:00 UTC. How can these be daily summed if there is only one map produced daily. I think the author's meant to say that the daily maps were summed over the extents of the ROIs. However this isn't a completely accurate description either as the TCCO values should be multiplied by the pixel area, correct? So in truth, per-pixel values of TCCO ($g.m^{-2}$) were multiplied by their respective pixel areas ($m^2$) and then summed over the extents of the ROI's. If correct, I'm not entirely sure what this yields. Having the units of Gg, it seems to me that this is the total CO contained by the atmosphere in the ROI's? It's not the excess CO above background due to biomass burning since both the model and the observations also account for the background. So I guess it's just the total amount of CO contained by the atmosphere in the ROI's, but I guess I would have expected to see this presented in ppm, not in absolute terms like Gg.

Also note that from Page 19, Line 382 onwards, the terms total column carbon monoxide (TCCO) and for a lack of a better term total carbon monoxide load (CO) are used interchangeably with interchangeable units of $g.m^{-2}$ and g, leading me to question whether the author's themselves understand what they've calculated. For example, on Page 19, Line 382, TCCO is presented as Gg, which up until this point, including Figure 9 on the previous page, TCCO was presented as $g.m^{-2}$. On Page 20, Figure 10, the total CO plotted in the left hand panels is presented as Gg, whereas Total Column CO plotted in the right hand panels is also presented in Gg. Finally, similar contradictions in terminology and units arise in Table 3 and Figure 11. I strongly suggest that the authors (i) define exactly what CO (in g) is supposed to represent, (ii) why the units of g instead of ppm are more appropriate, (iii) how CO in g differs from TCCO in $g\ m^{-2}$, and (iv) ensure that all remaining figures and tables are appropriately labelled with either CO or TCCO along with their respective units.

We thank the reviewer for his comments and recognise that our use of the terms TCCO and Total CO were not as clear as they could be. We have clarified these terms and added a description of how they are calculated. We have also amended figure captions, figures, and text to more clearly describe each variable in question.

As the reviewer rightly understands, Figure 10 shows the total amount of CO contained within the full domain or ROI on a given day (observed some time between 12:00 and 14:00 by S5P and an average of the CMAQ model output between 12:00 and 14:00 UTC). We chose to calculate total CO in a given area rather than to use concentrations for several reasons. Firstly, because the CMAQ model provides the model output variable Total Column CO, this can be directly compared to S5P TCCO (as it is in Figure 9). Secondly, since the model has a set number of layers and the S5P TCCO product simply gives a column total, assumptions about the atmospheric

vertical distribution of CO would otherwise be needed to calculate concentrations. Comparing CO area-aggregated totals removes this need.

Lines added at 426:

"The CMAQ model produces a TCCO output (mol.m$^{-2}$) that could be compared to Sentinel-5P TCCO (mol.m$^{-2}$) measurement data from June to August 2019. None of the S5P observations used in this comparison were those deployed in the $EC_{CO}^{b}$ derivation of Section **Error! Reference source not found.**. Prior to the intercomparison, both model and measurement datasets were converted to units of g.m$^{-2}$ via multiplication by the molecular mass of CO. S5P acquisitions over the model domain occur daily between 12:00 and 14:00 UTC, and the resulting TCCO retrievals were combined and compared with the mean CMAQ TCCO output from the same two-hour period."

Lines added 461:

"Area-aggregated CO totals (Gg) were calculated by multiplying both CMAQ and S5P TCCO by their 0.1° × 0.1° grid cell areas to obtain a daily summed total CO timeseries (between 12:00 and 14:00 UTC) for the full domain extent and within ROI1 and ROI2 (labelled in **Error! Reference source not found.**). These are shown in **Error! Reference source not found.**, along with direct comparisons of these daily area-aggregated estimates. These CO totals show that temporal patterns observed by S5P are well replicated by the CMAQ modelling driven by the FREM-derived CO emissions. This indicates that (i) temporal trends in active fires are being well captured in the SEVIRI FRP-PIXEL product and (ii) the meteorological fields of WRF, particularly wind, are representing the real conditions sufficiently well."

Technical Corrections:

1. Page 1, Title: Seems a bit asymmetrical to me. Suggest changing "geostationary" to "SEVIRI" or vice versa changing "Sentinal-5P TROPOMI" to "polar orbiting".

2. Page 1, Line 11: Likewise, to balance out "geostationary" in the first half of the sentence, I suggest changing "satellite observations of Total Column Carbon Monoxide (TCCO)" in the latter half of the sentence to "polar orbiting satellite observations of Total Column Carbon Monoxide (TCCO)."

3. Page 1, Line 18: Instead of "spanning 16 years" maybe state the actual range of years, 2004-2019 correct?

4. Page 1, Line 19: Suggest deleting "to derive CO emissions" from the end of the sentence.

5. Page 1, Line 20: Perhaps clarify that "per unit area" is "per unit burned area".

6. Page 1, Line 23: Here and elsewhere, I'm not sure that comparing outputs from the WRF-CMAQ chemical transport model with Sentinal-5P TROPOMI TCCO observations constitutes a "validation" of the FREM approach. It's an assessment, perhaps, but not a validation.

7. Page 2, Line 49: Suggest changing "actively burning fires" to "active fires" to coincide with the acronym "AF".

8. Page 2, Line 50: Suggest changing "relationship that relates a biome's fire radiative energy (FRE) to DMC totals coming from GFED" to "relationship between a biome's fire radiative energy (FRE) and DMC totals modelled in GFED".

9. Page 2, Lines 50-51: Suggest changing "The primary advantage over GFED" to "The primary advantage of GFAS".

10. Page 2, Line 53: Suggest changing "The main disadvantage is the fact that the relatively uncertain fuel load and combustion completeness assumptions, which introduce some of the most significant uncertainty to burned-area based fire emissions calculations, are incorporated into GFAS via this calibration" to "The main disadvantage is the fact that the relatively uncertain fuel load and combustion completeness assumptions, which introduce some of the most significant uncertainty to bottom-up fire emissions calculations, are also incorporated into GFAS via the calibration with GFED".

11. Page 2, Line 53: Granted the acronym FREM was defined in the abstract, but shouldn't it also be defined in body.

12. Page 3, Line 81: Maybe start a new paragraph?

13. Page 3, Line 91: Suggest changing first part of the sentence to read: "Using Equation 1 to translate between emission coefficients…"

14. Page 3, Line 93: Suggest moving "for example" to the beginning of the sentence.

15. Page 4, Lines 125-126: Suggest revising the latter half of the sentence to read "We also apply the cloud cover correction used in the LSA SAF Meteosat FRP-GRID product (Wooster et al., 2015), though the effect of this adjustment is limited due to sparse cloud cover during the African fire season.

16. Page 5, Line 144: Suggest replacing "temporally different" with "asynchronous".

17. Page 5, Line 145: This seems to imply that the 6 biomes were split into NHAF and SHAF to yield 12 relationships, which is not the case. I suggest rewording this sentence for clarification.

18. Page 5, Line 153: Do you mean 50% of the observed FRP or 50% or the observed number of AF pixels?

19. Page 6, Line 165: Were the AF pixels detected at a single SEVIRI timeslot or are they the cumulative collection of AF pixels detected from the start of the fire? Only two times are reported (11:24 and 11:30 UTC), which presumably correspond to VIIRS and Sentinel-5P, but what times were the AF pixels detected?

20. Page 6, Line 183: What is the acronym "MAIAC"?

21. Page 8, Line 200: Here the subscript represents the biome?

22. Page 9, Lines 208 and 211: The CO smoke emission coefficients were derived from the data shown in Figure 2, not Figure 1, correct?

23. Page 9, Line 208, Table 1: You set up the naming convention for the different inventories on lines 197-199 and then abandon this naming convention in Table 1. The first column reports the coefficients to be used in the FREM_bCO inventory and the second column reports the coefficients to be used in the FREM_bTPM inverntory, correct? It would be helpful if they were identified as such in the table caption and/or the table header.

24. Page 9, Line 221: Presumably the "s" in FREMs_bCO indicates a small fire correction, but this has not been mentioned up until this point.

25. Page 10, Line 230: Here the "s" in FREMs_bCO is first defined.

26. Page 11, Line 240: Here "small fire" is abbreviated "SF"?

27. Page 11, Line 244: This sentence does not make sense to me. To me, if you're discussing seasonal fluctuations, then the two emissions inventories would track on a monthly basis. If you're referring to interannual variability, then the two emissions inventories would capture the same large fire years and the same small fire years. Not really sure which of the two temporal scales you're trying to describe here.

28.  Page 11, Line 247: Something seems strange about the use of the parentheses here.

29. Page 11, Lines 253-254: How are the mean hourly CO emissions calculated from GFED4.1s? Using the monthly emissions and dividing by the number of hours in a month or using the daily/3 hourly fields? Not that it matters since total CO emissions from GFED4.1s should be conserved anyway, but in fairness I think it should be disclosed that it is possible to scale the monthly GFED emissions estimates to higher temporal resolutions.

30. Page 14, Figure 5: Suggest removing "emissions" from the panel titles in the figure. Also, the abbreviation should be CAR not CAF.

31. Page 14, Line 304: This is DMC per unit burned area, correct?

32. Page 15, Figure 6: Random "SF" for small fire again?

33. Page 15, Figure 7: Random "SF" for small fire again? As a stand-along figure caption there is no indication that this is dry matter consumed per unit burned area. Note also that between 11S and 13S and between 31E and 33E is not Angola. Figure 7 is a map of somewhere in western Zambia near to Malawi.

34. Page 16, Line 320: Again, can this really be considered a validation?

35. Page 16, Line 322: GFED4.1s is misspelled.

36. Page 16, 342-343: This sentence states that "Emission coefficients for all gas species used were calculated through the application of Equation 1 with CO as the reference species." Which other gas species were compared between the model runs and the FREM_bCO inventory? From here on forward I've only seen CO comparisons, which would not require the use of Equation 1. Moreover, this

is not a strictly true statement either as the CO emissions coefficients for the closed canopy forests are based on FEER-equivalent values, correct?  Also, it would be helpful to refer back to your naming convention. This is actually a part of the FREM_bCO emissions inventory, correct? Does this also include the SEVIRI small fire correction or not?

The CMAQ model requires other gas and aerosol species (beyond CO and TPM) to carry out the chemical and micro-physical processes of the model. Text has been added to make this more clear and to clarify that it is the FREMs_bCO emission inventory that is used as input (which includes a small fire correction).

37. Page 17, Line 365, Figure 8c: How are total carbon emissions being estimated? Via DMC? This is never described in the evaluation methodology, though I'm assuming that CO emissions are being divided by the CO emission factor to yield DMC which is then converted using the carbon fraction.

38. Page 18, Line 371: It seems a little late in the manuscript to start referring to Senitnel-5P as S5P. I suggest either introducing this abbreviation earlier or do not adopt it at all.

39. Page 18, Line 373: Presumably you are referring to Figure 8c?

40. Page 18, Line 380: It would be helpful if the figure caption stated that the mean monthly TCCO maps were generated for June, July and August using observations and model runs collected between 12:00 and 14:00 UTC.

41. Page 18, Line 380: So yes, according to panels (b), CMAQ was indeed fed with the FREMs_bCO emissions inventory. It would be helpful if this was stated somewhere in the methodology. See Comment 36.

---

## Author Comment (AC2)

**Anonymous Reviewer #1**

We thank this reviewer for their input and comments, which raise important and useful points, we have incorporated their suggestions and feel the manuscript has been improved as a result. Comments that have been addressed are highlighted in green.

Line 61: "The FEER and FREM approaches derive landscape fire emissions estimates directly from EO-derived FRE measures, removing the step requiring calculation of DMC and thus the uncertainties inherent in the calculation." While this is a true statement, it does not mention the trade off in uncertainties from taking a different approach. There are uncertainties inherent in the estimation of the smoke emissions coefficient. If the authors would like to assert that the top-down approach has lower overall uncertainty, they will need to support that in the text.

This is certainly a true and valid point and we have included text to make this more clear. Lines added at line 76:

"Although uncertainties in the DMC conversion step are removed in these top-down approaches, other uncertainties are introduced - primarily from uncertainties in the satellite-derived datasets, and in the case of $EC_{TPM}^b$, the use of the mass extinction coefficient, $\beta_e$, used in the conversion of AOD to TPM"

Line 76: While the geostationary satellites provide higher temporal resolution, and it is explained why this is desirable, they also provide lower spatial resolution. Please explain the benefits and tradeoffs.

Also a very good point which was perhaps not as clearly stated as it should have been, Lines added 86:

"A drawback of using geostationary AF data is that, at present, operational geostationary satellites have a lower spatial resolution than do polar-orbiting sensors, resulting in the under-detection of 'small' or low-FRP fires. This 'missing' contribution to the FRE can be accounted for using a so-called 'small fire correction' (discussed in more detail in Section 2.1)."

Lines 75, 96, 277, 292, 293, 307, 466, 558: Take out the word "far" entirely or replace it with something quantitative. This is overly qualitative and the reader should decide what is or isn't "far greater," "far higher," "far more consistent," etc.

These have been removed

Line 123: I would like to see at least a short description of the "small fire adjustment" factor used in the analysis, beyond just the reference. It appears to be important. Table 2 indicates a 50% difference with our without this correction. This makes it a core part of the method that should be discussed.

Thank you for this input. Lines have been added discussing in more detail this adjustment. Line 134:

"Region-specific mean 'small fire' scaling factors were derived from comparisons of coincident and co-located aggregated FRP from the MODIS and SEVIRI active fire products. These were determined to be 1.67 and 1.46 for Northern and Southern Hemisphere Africa (NHAF and SHAF) respectively, and following Mota and Wooster (2018) these factors were applied to account for the average amount of FRP coming from fires burning below the SEVIRI sensor's minimum FRP detection limit."

Line 173: Why was OLS used over ODR. I appreciate that the original work was recalculated using OLS for consistency, but was there a reason for the switch? The new values are 14% lower. Is that a better estimate? No justification was given for the switch so I'm not sure which version I should prefer.

Thank you for this input, this choice could have been explained in more detail and text has been added to discuss the reasoning for this methodological choice. Line 198:

"Matchup data for each 'fire biome' are shown in Error! Reference source not found. and were used to derive the set of biome-dependent CO smoke emission coefficients [$EC_{CO}^b$] listed in **Error! Reference source not found.**. Zero-intercept ordinary least squares (OLS) regression was used for this, rather than the orthogonal distance regression (ODR) used by Nguyen and Wooster (2020) during derivation of TPM smoke emission coefficients [$EC_{TPM}^b$]. OLS was used in this work for two main reasons. Firstly, although the ODR method considers the uncertainty in each of the variables, these uncertainties are themselves rather poorly constrained, with the known uncertainties only representing part of the total uncertainty sources. There are contributions to the uncertainty of FRP that are not quantifiable, for example due to variations in the amount of interception of a fire's FRP signal by any overlying tree canopy. We therefore deemed use of a regression method in which the slope is strongly driven by datapoint uncertainty to be unsuitable for use. Secondly, weighting based on uncertainty often resulted in undue weight being given to high value points (e.g. match-ups with high FRE and high plume-species amounts) due to them typically having lower relative uncertainties (see Wooster et al., 2015). Due to their typically being very few high value datapoints in each 'fire biome' due to the heavy tailed nature of fire size distribution (Freeborn et al., 2009), these few large fires were potentially being too strongly weighted in the resulting calculation of $EC_x^b$. For these reasons, we opted to use OLS regression, and to ensure a consistent methodology for emission coefficient derivation we also applied the same approach to the Nguyen and Wooster (2020) dataset to re-derive their $EC_{TPM}^b$ values using OLS regression (see **Error! Reference source not found.**). The updated $EC_{TPM}^b$ for closed canopy forest, managed land, grassland, shrubland, low-woodland savanna and high-woodland savanna are 26.07 g.MJ$^{-1}$, 12.23 g.MJ$^{-1}$, 9.39 g.MJ$^{-1}$, 9.88 g.MJ$^{-1}$, 10.65 g.MJ$^{-1}$, and 14.18 g.MJ$^{-1}$ respectively. On average these new values are 14% lower than those reported in Nguyen and Wooster (2020) derived via ODR, and are the ones referred to and used hereafter. The WRF-CMAQ model-based approach to evaluating our final CO emissions rates and totals in Section **Error! Reference source not found.** was also used to carry out an analogous evaluation of the TPM emissions

generated from the updated $EC_{TPM}^b$ values of **Error! Reference source not found.** (see **Error! Reference source not found.**).”

Line 187: “TROPOMI CO plumes in the closed canopy forest biome were not sufficiently distinct from the background in this biome.” Is this a shortcoming of TROPOMI CO, the method, or just particular to the region? In some regions, closed canopy forests are the primary source of biomass burning emissions. It would be good to get a discussion on how applicable this approach is to other parts of the world.

This is a very helpful and valid line of questioning. A discussion of this issue has now been added at Line 219:

“For each of the six fire biomes at least 12 matchup fires were identified for derivation of $EC_{CO}^b$, apart from for closed canopy forest. Tropical evergreen forests (the primary type of closed canopy forests in tropical regions) are generally not very susceptible to fire, except during periods of extreme drought, due to the high humidity and low windspeed within the dense forest canopy and the limited amount of surface fuel available due to rapid decomposition of surface litter in these environments (Marengo et al., 2011; Tomasella et al., 2013). Fires in such tropical forests are most often the result of human land-clearing activity and are typically small in size, unless heavy machinery is involved in land clearing (Van Leeuwen et al., 2014). Furthermore, FRP observations in closed canopy forest can be affected by tree canopy interception of surface emitted FRP (Roberts et al., 2018). These factors result in a lower number of observable and identifiable fire match-ups for tropical closed canopy forest areas. Smaller fire sizes and fewer match-up fires being acquired in closed canopy forests areas relative to other biomes was also observed during previous applications of the FREM approach (Mota & Wooster, 2018; Nguyen & Wooster 2020) and the FEER approach (Ichoku & Ellison, 2014), even when using 7 years' worth of MODIS FRP and AOD data in the latter case. In this work the ability to identify small fires in closed canopy forest is further limited by i) the spatial resolution of the S5P TCCO observations which are at least 5 times lower resolution than the 1 km AOD product used in Nguyen & Wooster (2020) and ii) the limited availability of the S5P trace gas products which only became operational form mid-2018. An increased timeseries of S5P data and the exploitation of machine learning methods such as object recognition may aid in identifying a greater number of plumes in closed canopy forest - and this the subject of ongoing work.

Due to the FEER emission inventory exploiting a far larger dataset from which to identify fire match-ups (7 years of MODIS FRP and AOD) and it obtaining many more fire match-ups in tropical closed canopy forest (Ichoku and Ellison, 2014) we instead derive $EC_{CO}^b$ for closed canopy forest from the 'FEER-equivalent' value. The method used to derive this is detailed in Nguyen and Wooster (2020), and essentially involves aggregating the FEER $C_e^{TPM}$ emission coefficients of Ichoku and Ellison (2014) (https://feer.gsfc.nasa.gov/data/emissions/) to the relevant fire biome. Equation 1 was then applied to obtain a FEER-equivalent $EC_{CO}^b$ , which was calculated as 156.7 g.MJ$^{-1}$ for the closed canopy forest fire biome. We generated FEER-equivalent $EC_{CO}^b$ for each of the other five fire biomes to compare these to our directly derived $EC_{CO}^b$ values, and found agreement within ±34% (see **Error! Reference source not found.**), somewhat justifying our use of the FEER-equivalent value in the closed canopy forest biome where a directly derived $EC_{CO}^b$ value was not achieved. Further, Nguyen and Wooster(2020) showed that mean monthly FRE contribution from fires in closed canopy forests does

not exceed 10% of the total monthly FRE coming from African fires, and thus its total is of relatively low importance to continental-scale CO emissions totals."

Table 1: I'm not sure I understand the need for $C^{CO}_e$ calculated via $C^{TPM}_e$. What is the reason for showing this?

This has been included to demonstrate the difference in the predicted CO emissions that result from using a different reference species. This difference results from the combined contribution of the top-down derivation, which uses a different reference species satellite product (MODIS MCD19A2 AOD), and the application of Equation 1 which uses emission factors which are themselves not well constrained. Material has been added to the main text and caption to highlight purpose of this this.

Lines 245-249: Does this discrepancy between FRP and CO emissions timing suggest that the relationship depends on the type of combustion? Are long-lived smoldering fires prevalent in this region? Do the authors have any hypothesis for this?

This is an interesting point and we thank the reviewer for noting this, we have added lines discussion the proposed reasons for this. Lines 319:

"However, as Mota and Wooster (2018) and Nguyen and Wooster (2020) noted for TPM emissions, the FREM methodology often predicts a slightly earlier peak in annual emissions in SHAF compared to GFED. This shifted peak agrees with findings showing that polar-orbiting based FRP measures also seem to peak in SHAF a month or so earlier than do BA measures. This appears, for example, in the work of Zheng et al. (2018) who compared GFED BA with GFAS FRP and who suggest that measured CO emissions actually lag BA derived CO emissions in Africa, based on MOPITT CO observations. They attribute this lag to a shift from flaming to smoldering combustion over the continent which is not accounted in the emissions factors applied in the GFED calculations."

Figure 9. For intercomparison between the figures. it would be useful to have outlines of the ROIs on these maps. This would be more helpful to me than the city names, which are not included on Figure 8.

Indeed, the ROIs would be more valuable here and have been added to Figure 9. Site names have been retained for ease of comparisons between the CO and AOD spatial distribution results.

Technical/Grammatical Comments:

Line 30: Recommend removing allusions to a potential future product in the abstract.

We feel this is not inappropriate as the technical development of this product is currently underway, however this has been re-worded.

Line 58: Remove "fully"

Line 62: Define "EO"

Line 211: "Figure 1" Is this a typo? I don't see how Figure 1 shows this at all.

Line 326: Remove parenthetical. Those details are given below on line 348.

Line 456: maybe --> may be

Line 598: will available --> will be available

---

## Author Comment (AC3)

Anonymous reviewer #2

We thank the reviewer for these helpful insights, we have tried to address the reviewer's comments and questions to the best of our ability and feel this has added clarity to the manuscript. With respect to questions about the relevance of this manuscript to ACP - we feel that the work presented is firmly within the subject area of the journal ('The main subject areas comprise atmospheric modelling, field measurements, remote sensing,...'). Similar works describing both top-down and bottom-up global biomass burning emissions inventories have been previously published in ACP (e.g. Van der Werf et al., 2010; Ichoku & Ellison, 2014) along with other remote-sensing based approaches to estimating biomass burning emissions (e.g. Chang and Song, 2010; Mao et al., 2014; Wu et al., 2018).

The paper presents the retrieval of carbon monoxide emissions over Africa from a 16-year archive of geostationary satellite images, with several auxiliary data derived from other satellite instruments, such as Tropomi. It's a dense technical document, which certainly represents a huge amount of work. The reader of the ACP might be surprised by its contents, because in the portfolio of the EGU journals, it would fit much better with the AMT ("advances in remote sensing") than with the ACP ("studies investigating the Earth's atmosphere and the underlying chemical and physical processes"). That aspect aside, it is full of technical processing details and acronyms that do not make it easy to read, but I still did not find some of the key information I was looking for, leaving me to wonder about the quality of other information about which I know less:

What Tropomi data was used? NRTI, OFFL? The document states that the 2020 data was used at 7 km resolution as for 2018, but the Tropomi data was at 5.5 km resolution along the track during the second period: how was the degradation of the resolution made? Was data with variable "qa_value" less than 1 correctly discarded? How were the stripes of CO data in the direction of flight treated for the two time periods? Can the selection of Tropomi data bias the results towards certain types of fires (with low aerosols optical thicknesses for example)?

We thank the reviewer for this comment, there are several elements and we have separated their questions for ease of addressing them.

What Tropomi data was used? NRTI, OFFL? The document states that the 2020 data was used at 7 km resolution as for 2018, but the Tropomi data was

This work uses Sentinel-5P TROPOMI data from 01/07/2019 to 31/12/2019 plus that from the entirety of 2020. The NRTI_L2_CO product is typically only available on the Copernicus datahub (https://s5phub.copernicus.eu/dhus/#/home) for 24 hours after the current time. Since we used many months of observations we rather used the OFFL product. We have added this clarification to the methodology section and added text detailing the change in the along-track resolution of the S5P product in 2019 and how this was dealt in when calculating Total plume CO.

Lines 117:

"The offline (OFFL) S5P total column carbon monoxide (TCCO) product used in this work can be downloaded from Sentinel-5P Pre-Operations Data Hub (https://scihub.copernicus.eu/) and has a spatial resolution of  7 × 7 km until August 2019 after which the along-track resolution increased to 5.5 km."

Line 178:

"For each fire in the final match-up set we calculated i) the total CO [g] contained in the plume from the S5P TCCO retrievals, and ii) the total FRE [MJ] released by the fire from Meteosat FRP-PIXEL data – integrating FRP from the time of the fires first AF pixel detection on that day to the moment of the S5P overpass. The minimum TCCO value ($mol.m^{-2}$) from each plume buffer was taken as the appropriate CO 'background' value for that plume and subtracted from all plume pixel TCCO values. The resulting 'excess' TCCO pixel values were then converted to units of grammes by multiplying by the molecular weight of CO and by the pixels' area calculated from their geographic corner coordinates (thus accounting for the change in the along-track spatial resolution of S5P data in 2019 and any view-angle dependent pixel area growth). Summing these values across all plume pixels of a fire then yielded the total amount of fire-emitted CO in that plume, which was compared to the total amount of FRE the fire generated over the time that it released that CO. Each match-up fire FRE and total CO pair constitutes one datapoint on the relevant 'fire biome' scatterplot of **Error! Reference source not found.**. FRE and Total CO uncertainties were calculated from the uncertainty values provided within the FRP-PIXEL product and the TCCO product respectively, these are also plotted in the form of error bars on Error! Reference source not found.."

Thank you for this question. The *qa_value* in the S5P TCCO product gives an indication of whether a S5P pixel passes tests for contamination by mid-level and high-level cloud. Our methodology does not eliminate TCCO pixels with qa_values less than 0.5 (as recommended in the product metadata) because the presence of cloud is already determined and delt with through the use of the two other (far higher spatial resolution) satellite products used in tandem

with the S5P product. Firstly, we use the 3 km spatial resolution SEVIRI FRP-PIXEL Quality Product (associated with the FRP product) to inspect and filter out any potential match-up fires that might be contaminated with cloud during the duration of the fire. These 15 min temporal resolution data ensure that no clouds were present over the location of the fire at any point during it's lifetime. Secondly, we use VIIRS RGB imagery (375 m spatial resolution at nadir) acquired within 3.5 minutes of the S5P data to ensure the S5P observations are cloud free. For these reasons we do not need to use the S5P CO qa_value.

How were the stripes of CO data in the direction of flight treated for the two time periods?

We are certainly aware of the Fixed Mask De-striping (FMD) method proposed by Borsdorff et al. (2019) for Sentinel-5P data. Indeed, the FMD is now applied in the Sentinel-5P OFFL operational product from July 2021 onwards, but is not applied to the same product prior to that date. You can create the FMD method yourself and apply it to such data – however we chose not to apply it to the retrospective dataset analysed in the main text. This choice was made based on our understanding of the FMD method, its "validation" in Borsdorff et al. (2019) and the situations under which it was evaluated therein,  plus the results of our own tests of the impact that applying the de-striping had on fire plumes.

The FMD "correction" is essentially an adjustment of the TCCO values delivered at each pixel position across TROPOMI swath, based on adding or subtracting a constant value at each location. The adjustment is thus irrespective of the actual TCCO value in each pixel in the swath, and constant at each swath pixel position. In Borsdorff et al. (2019) an evaluation of this de-striping procedure was carried out by comparing daily averaged de-striped S5P TCCO values against a TCCO measurement made by one of the TCCON ground stations. Whilst the TCCON station was a single point observation, the S5P data were averaged over a colocation radius of 50 km and typically the CO values varied rather mildly across this 50 km region.

The temporal and spatial scales of the CO variations seen in these validation locations are clearly very different from the application of the data in our work, where very strong spatial CO gradients exist at the location of the fire plumes. Although allusions to the impact of the (de)striping at smaller scales are made in Borsdorff et al. (2019), no solid analysis is presented showing the impact of the striping, or de-striping on such phenomena. We tested this with our data, however, and observed that in the case of smoke plumes (which are represented in the TROPOMI product by small groupings of pixels with high TCCO concentrations) the striping clearly has less impact on the high TCCO values in the plume than it does on the low TCCO pixels of the plume

background. This can be seen in the below plots (Figure 1) that show two different regions containing smoke plumes on 23-07-2022 at 11:01 UTC. The plumes are circled in red, and the image comparisons (top vs bottom images) show that over these relatively small areas, low "background" TCCO pixels are more affected as a percentage of their TCCO by both the striping and the FMD 'de-striping' procedure than are the high "plume" TCCO pixels. It is not at all clear here, nor in the original Borsdorff et al. (2019), that a constant de-striping correction such as the FMD approach is appropriate to apply in an area where TCCO concentrations have such a high spatial gradient between pixels. Furthermore, the FMD method does not actually remove all the stripes, neither here nor in the original examples of Borsdorff et al. (2019). Hence, we have chosen not to apply the FMD procedure until more quantitative evidence of it's appropriateness to such high CO gradient regions is published. For completeness however, we have applied it to the data from which we calculate the CO emission coefficients ($EC_{CO}^b$) for each fire biome – and included these results in a new Appendix along with an explanation. The main manuscript still includes our CO emission coefficients derived with the standard CO product, whilst the Appendix includes the values calculated from the same data but with the FMD procedure applied (it also shows the % difference between these and the ones in the main manuscript – see Table 1 (below)). The FMD procedure has a greater impact on low CO value "background pixels" than on the high CO "plume" pixels - as observed in Figure 1 below – and the maximum smoke emission coefficient change is 10% when FMD is applied compared to when it is not (see Table 1) – and in all cases $EC_{CO}^b$ is slightly lower with destriping applied.

[Figure]

[Figure]

Figure 1: Comparison of standard and Fixed Mask De-striped S5P CO data of two fire plume regions (Region 1 at left and Region 2 at right).

Figure 2: CO Emissions coefficients derived with FMD applied to S5P CO data.

Table 1: Comparison of smoke emission coefficients with and without the FMD de-striping procedure applied.

| | S5P without FMD applied | S5P with FMD applied | Percentage difference (%) |
|---|---|---|---|
| Grassland | 75.51 | 68.94 | -8.7 |
| Shrubland | 81.07 | 76.16 | -6.1 |
| Managed land | 88.35 | 82.91 | -6.2 |
| High-woodland savanna | 81.85 | 79.99 | -2.3 |
| Low-woodland savanna | 85.49 | 77.04 | -9.9 |

Can the selection of Tropomi data bias the results towards certain types of fires (with low aerosols optical thicknesses for example)?

We are somewhat unclear what the question is aiming at here. We think it is related to the fact the TROPOMI products mask out data with very high aerosols, so therefore our dataset might not include fires that produced such characteristics. We don't expect such biases on the basis of what we see – which includes fires with significant aerosol plumes. What we do have is a bias towards larger and longer-lived fires (No shorter than 30 minutes with 70% of matchups more than 2 hours), as very small or short-duration fires will not generate the sort of data record necessary for the matchups. However, since we find a linear relationship across lower and higher FRP fires in the matchup scatterplots (see Figure 2 in the manuscript) we have no evidence for the introduction of bias.

How is the plume buffer computed (I cannot guess it from Fig 1)? It is easy to bias the "background" estimation high or low with a slight change in the buffer definition, in particular due to the unstable choice of the minimum value.

The fire match-ups and plume polygons were identified manually and the buffer around each plume was selected such that i) it visibly incapsulates the high CO pixels of the plume and ii) gives several pixels' worth of close-to-background values surrounding the plume. Defining a buffer of a set pixel width is not appropriate as this has the potential to result in contamination from neighbouring plumes, which can be close by during periods of greater fire activity. In all previous top-down approaches that derive emissions coefficients from a dataset of fire match-ups, no single or consistent method

has been used for defining the background level of the variable of interest (previously this was always AOD). For example, Ichoku & Ellison (2014) and Fisher et al. (2020) define the AOD background as the value of a pixel or group of pixels up-wind of each plume, while Mota & Wooster (2018) use a large area average (500 km$^2$) of AOD surrounding the plume. In the previous version of FREM (Nguyen & Wooster, 2020) and in the current work, the use of a percentile threshold approach to define the ambient background was explored - this method relies on defining the lowest, say, 5% of pixels within the plume polygon to be the background CO or AOD value while the rest are defined 'plume' pixels. However, this approach was found to be more unstable than a minimum pixel approach because a percentile approach means that an individual pixel has a different weight contribution to the background value depending on the size of the plume. The minimum pixel value was shown to be the more stable of the two for this work.

What is the time scale of the FRP temporal integration? Can some of the CO have been oxidized during that time?

The temporal integration of these fires is in the order of hours, the average integration time of the match-ups was 3.4 hours and 80% of the plumes had integration times below 5 hours. With a typical atmospheric lifetime of 2 months, significant oxidation of the plume CO is highly unlikely in the time between the first AF detection and the time of the S5P overpass.

How are the boundary conditions treated in the model simulations?

Both Initial and Boundary conditions for the meteorology (WRF) are taken from the NCEP FNL global model which has a grid resolution of 0.25° × 0.25° at 26 levels and 6 hour timesteps. Initial and boundary conditions for the chemistry (CMAQ) are taken from the WACCM global chemistry model having 0.9° × 1.25° spatial resolution, 88 levels, and 3 hour timesteps. These details are mentioned briefly in Section 4.1 and further model details are listed in full in Appendix B.

Apart from the uncertainty of the slope (and not the "error" of the slope as wrongly put in the legend, in fact without any definition of the uncertainty variable used), there is no notion of uncertainty in the product shown here.

We thank the reviewer for this valid and insightful comment. Both the FRP and CO satellite observations have associated uncertainties, and the uncertainties in total

plume FRE and CO were calculated from these data but were previously not plotted on Figure 2 of the manuscript. These uncertainty measures have now been included in the form of error bars on Figure 2. These uncertainties were not included in the original manuscript as they represent only part of the uncertainty associated with the total fire FRE and CO i.e. the uncertainty inherent in the mathematics to calculate these variables and the radiometric measurements of the satellite. There are contributors to the uncertainty in the match-up FRE and CO that are not included and not quantifiable in the same manner. For example, the plotted uncertainty doesn't include uncertainty in FRE associated with how much of the fire emitted energy is intercepted by tree cover. Weighting the derivation of the $EC_{CO}^b$ values with data whose weights depend on incomplete and poorly constrained uncertainties was deemed inappropriate, and thus ODR regression (which does just that) was rejected in favour of OLS regression. A discussion has been added on this point (see below).

Lines added regarding OLS selection. Line 198: following

"Matchup data for each 'fire biome' are shown in Error! Reference source not found. and were used to derive the set of biome-dependent CO smoke emission coefficients [$EC_{CO}^b$] listed in **Error! Reference source not found.**. Zero-intercept ordinary least squares (OLS) regression was used for this, rather than the orthogonal distance regression (ODR) used by Nguyen and Wooster (2020) during derivation of TPM smoke emission coefficients [$EC_{TPM}^b$]. OLS was used in this work for two main reasons. Firstly, although the ODR method considers the uncertainty in each of the variables, these uncertainties are themselves rather poorly constrained, with the known uncertainties only representing part of the total uncertainty sources. There are contributions to the uncertainty of FRP that are not quantifiable, for example due to variations in the amount of interception of a fire's FRP signal by any overlying tree canopy. We therefore deemed use of a regression method in which the slope is strongly driven by datapoint uncertainty to be unsuitable for use. Secondly, weighting based on uncertainty often resulted in undue weight being given to high value points (e.g. match-ups with high FRE and high plume-species amounts) due to them typically having lower relative uncertainties (see Wooster et al., 2015). Due to their typically being very few high value datapoints in each 'fire biome' due to the heavy tailed nature of fire size distribution (Freeborn et al., 2009), these few large fires were potentially being too strongly weighted in the resulting calculation of $EC_x^b$. For these reasons, we opted to use OLS regression, and to ensure a consistent methodology for emission coefficient derivation we also applied the same approach to the Nguyen and Wooster (2020) dataset to re-derive their $EC_{TPM}^b$ values using OLS regression (see **Error! Reference source not found.**). The updated $EC_{TPM}^b$ for closed canopy forest, managed land, grassland, shrubland, low-woodland savanna and high-woodland savanna are 26.07 g.MJ$^{-1}$, 12.23 g.MJ$^{-1}$, 9.39 g.MJ$^{-1}$, 9.88 g.MJ$^{-1}$, 10.65 g.MJ$^{-1}$, and 14.18 g.MJ$^{-1}$ respectively. On average these new values are 14% lower than those reported in Nguyen and Wooster (2020) derived via ODR, and are the ones referred to and used hereafter. The WRF-CMAQ model-based approach to evaluating our final CO emissions rates and totals in Section **Error! Reference source not found.** was also used to carry out an analogous evaluation of the TPM emissions generated from the updated $EC_{TPM}^b$ values of **Error! Reference source not found.** (see **Error! Reference source not found.**)."

Just by looking at table 1, I understand that the 5 key numbers derived from the Tropomi data are very close to each other: are their differences statistically significant? –

This is a very valuable comment and we thank the reviewer for this input. We have taken pairs of fire biomes and assessed whether their derived $EC_{CO}^b$ values are statistically significantly different. We have also carried out the same analysis for the updated $EC_{TPM}^b$ values of Appendix A. These results have now been added in two tables and a discussion included in the main text.

Line 255:

"Tests were carried out to determine the statistical similarity of the $EC_{CO}^b$ values shown in **Error! Reference source not found.** for each fire biome. Results are shown in **Error! Reference source not found.**, and indicate that only one pair of biomes (grassland and managed land) have CO emissions coefficients ($EC_{CO}^b$) that are statistically distinct at the 95% confidence interval. Two additional fire biome pairs are statistically distinct at the 85% confidence interval (grassland and low-woodland savanna; managed land and shrubland). This analysis indicates that based on the current match-up dataset, there is relatively little inter-biome difference in the relationship between the amount of thermal radiant energy (i.e. FRE) emitted by a fire and the amount of CO emitted by the same fire. However, smoke emission factor databases (e.g. Akagi et al., 2011; Andreae, 2019) indicate that $EF_{CO}$ values do differ significantly between certain biomes. The updated FREM TPM emissions coefficients ($EC_{TPM}^b$) of Appendix A were similarly tested for statistically significant differences, and the emission coefficients generated from this (much larger) match-up dataset (primarily due to the higher spatial resolution of the MCD19A1 AOD product allowing more plumes to be identified) were found overall to be more statistically significantly different than those of CO. Of the fifteen biome pairs only four did not have $EC_{TPM}^b$ that were statistically significantly different at a 95% confidence threshold. This reduces to 2 pairs at the 85% confidence limit. It may be the smaller sample size in the case of the CO data is not allowing the biome-driven differences to manifest themselves in a significant enough way - and limitations to producing a larger sample size in this study have previously been discussed along with the future research work needed to address this key issue."

An error budget for the complete processing chain is surprisingly lacking to support, for example, the discussion of p. 12-13. As of now, this discussion looks like mere speculation.

One of the largest challenges in the assessment of biomass burning inventories of any kind (top-down or bottom-up) is the lack of uncertainties – and this includes for example the GFAS and GFED fire emissions inventories that are by far the most widely used of any worldwide. There is also at present no feasible way to directly compare emissions estimates with an absolute ground truth. As a result, the

methods used to evaluate the validity of a fire emission inventory are typically; sense-check comparisons to other fire emissions inventories; large spatio-temporal aggregation of emissions that are then compared to satellite observations of gas or aerosols for specific cases; validation of the input data, i.e. FRP or Burned Area; or an in-direct evaluation through atmospheric modelling and comparison of the output fields to atmospheric observations (as is carried out in Section 4 of this work). The discussion to which the reviewer refers simply serves the purpose of highlighting what previous studies have shown about the current and previous versions of GFED and how our results compare with those. Perhaps our suggestions referring to how the results of this work support the conclusions of previous work may be overstated, so we have re-worded these points. The bulk of the discussion, however, simply makes reference to other studies, or states the results of this work.

In addition, two minor aspects should be addressed:

- The title contains a typo ("Sentinal") and a not-so-common acronym (FRP)

  The title has been changed

- The term "measure" used in many places about remote sensing is not appropriate (see the BIPM metrology definitions).

  Remote sensing does include a measurement – of light of certain wavelengths. Therefore, we feel it is appropriate to use the term – and also just for readability we have included it. Where possible we say "estimated" or "derived" since for example CO column amounts are derived from Sentinel-5P observations, rather than CO being directly measured.

---

## Author Response (AR2)

Reviewer #2 comments:

The authors have clarified the points that I raised in a convincing way, except for the following ones:

- The authors explain that they do not need the qa_value filter because they have other means to detect clouds in the scenes. Their use of ancillary data is commendable, but they cannot ignore the flags that come with the retrievals. For example, if the retrieval assumed there is a cloud, its averaging kernel will likely peak higher in the atmosphere and will less represent a total column. Authors should remove all data with qa_value less than 1 to avoid misinterpretation.

We are happy to hear that Reviewer #2 is convinced by our replies, and appreciate the expansion on the point originally raised. As suggested by the reviewer on the remaining point, we have now used the qa_value in the Sentinel-5P CO product to threshold the data used. The reviewer suggests removing all data with qa_value < 1.0 ; though we believe this rather conservative threshold is both unnecessary and impractical – based on the detail we provide below. We have rather removed all data with qa_value < 0.5 from all match-up plumes in our dataset, based on the recommendations and information in the Sentinel-5P CO User Manual and Sentinel-5P CO Product Readme document on how to use this qa_value.

**Detail of threshold selection:** A discussion of the sensitivity of the vertical averaging kernels used in the TROPOMI Total column CO retrieval can be found in Borsdorff et al. (2014). The Figure below is taken from that work and shows the difference in the mean global averaging kernel on a single day (10$^{th}$ November 2017) of observations with differing qa_values . We can see that at the typical altitudes at which the large smoke plumes from the types of fires used in our matchup process are observed (typically 800 m to 4000 m altitude), the weighting of the averaging kernel [$a_{col}$] applied to pixels containing mid-level cloud (i.e. pixels with qa_values > 0.5 but < 1.0; black line) differs by no more than 0.1 from the weighting of the averaging kernel applied to clear-sky pixels (yellow line). The weighting applied to pixels with qa_value = 1.0 at these altitudes is also well within the standard deviation (horizontal bars) of the that applied to pixels with qa_values > 0.5. In fact, out of all altitudes, the sensitivity to total column CO is closest between pixels with qa_values > 0.5 & < 1.0, and pixels with qa_value = 1.0 at the altitudes which the bulk of a fires smoke plume is likely to be observed at after several hours of burning (approximately 1500-2000 m). Indeed, Borsdorff et al. (2014) state that in remote regions, such as the locations of the plumes in our matchup dataset, the error introduced by the choice of averaging kernel is comparable between clear-sky observations and observations containing mid-level cloud (qa_value > 0.5 but < 1).

[Figure]

**Figure 1.** TROPOMI CO total column averaging kernels for 10 November 2017. The global average is shown for three different categories of cloudiness strict cloud clearing (black), clear-sky equivalent (yellow), and high optical thick clouds (blue). The standard deviation is indicated as error bars.

It should also be noted that the qa_values provided in the S5P CO product have not been validated in conditions anywhere near comparable to the application in which we have used the product. S5P CO data have been evaluated using the TCCON and NDACC ground-based solar FTIR monitoring networks, with this conducted by averaging the S5P CO product over 50km² areas around each ground measurement site and comparing the total column CO values. None of these sites were located in Africa, and the validation was based on large area averaging of ambient-type total column CO data from clear-sky conditions (Borsdorff et al., 2014). Our matchup dataset features total column CO observations over far smaller areas with extremely elevated CO and strong spatial CO gradients. The application of the qa_values applied in Borsdorff et al. (2014) is therefore unlikely to be fully representative of our application, and we hence treat the qa_values assigned in the S5P data with caution. Therefore, we set a qa_value threshold of 0.5, with this slightly more lenient threshold being especially appropriate as we have also used the extremely cloud-sensitive LSA SAF Meteosat cloud mask and the VIIRS imagery (which has a 750 m spatial resolution) of the same time as the S5P overpass to already remove any cloudy matchups from our candidate matchup plume dataset. We therefore confirm via the qa_value thresholding and the image analysis that no matchups that are cloudy exist in our matchup plume dataset. We trust this new use of the qa_value and our confirmation of cloud free data now satisfies the reviewer.

Having applied the qa_value threshold as detailed above, the resultant emissions coefficients ($EC_{CO}^{b}$) are shown below in Figure 2. These $EC_{CO}^{b}$ values are summarised in Table 1 along with the $EC_{CO}^{b}$ values derived when the previously discussed Fixed Mask De-striping (FMD) is applied, and the respective percentage difference between these and the original $EC_{CO}^{b}$ values presented in the original manuscript. The qa_values > 0.5 thresholding results in an average $EC_{CO}^{b}$ difference of only

2.4% across all five biomes (maximum difference is 5.9%) demonstrating that the impact of the mid-level cloud on the retrieval of CO in our study is rather minimal and well within the uncertainty bounds of the $EC_{CO}^b$ values. As discussed above, it is unclear from the validation carried out by Borsdorff et al. (2014) on the S5P CO product whether the qa_values defined for each pixel are really appropriate for use in the extreme conditions in which we utilise the S5P CO data. Combined with the apparent minimal impact that the removal of qa_values < 0.5 has on our calculated $EC_{CO}^b$ values, we have included the coefficients with the qa_value threshold applied in the Appendix - for the benefit of those that may prefer to use these values and interested to understand. We have also added a condensed explanation of the above in the main manuscript text and in the Appendix.

[Figure]

| | S5P without FMD applied | S5P with FMD applied | Percentage difference (%) | S5P excluding qa_value < 0.5 | Percentage difference (%) |
|---|---|---|---|---|---|
| Grassland | 75.51 | 68.94 | -8.7 | 73.49 | 2.7 |
| Shrubland | 81.07 | 76.16 | -6.1 | 82.26 | -1.5 |

| | | | | | |
|---|---|---|---|---|---|
| *Managed land* | 88.35 | 82.91 | -6.2 | 87.63 | 0.8 |
| *High-woodland savanna* | 81.85 | 79.99 | -2.3 | 77.03 | 5.9 |
| *Low-woodland savanna* | 85.49 | 77.04 | -9.9 | 86.62 | -1.3 |

Reference

Borsdorff, T., Aan De Brugh, J., Hu, H., Hasekamp, O., Sussmann, R., Rettinger, M., Hase, F., Gross, J., Schneider, M., Garcia, O., Stremme, W., Grutter, M., Feist, Di. G., Arnold, S. G., de Mazière, M., Kumar Sha, M., Pollard, D. F., Kiel, M., Roehl, C., … Landgraf, J. (2018). Mapping carbon monoxide pollution from space down to city scales with daily global coverage. *Atmospheric Measurement Techniques*, *11*(10), 5507–5518. https://doi.org/10.5194/amt-11-5507-2018

- "It is rather more likely that the sample size of the current work is not sufficiently large to enable statistically distinct $ECCOb$ values to be derived". I do not understand the argument because a small sample size will unlikely bring the values close to each other (ie by accident) as it happens here.

We thank the author for their comment. This statement was certainly unclear and in need of further expansion, which we provide below and in the main text.

Each of the fire-plume matchups in our dataset are classified into one of the 6 distinct 'fire biome' classes, based on the location of the active fire pixels that make up the fire. As such, in most cases a smoke plume is not 100% generated by fire that burns vegetation of a single biome – because the active fire pixels can be from more than one biome. We therefore use a filter to ensure that the majority (>50%) of the active fire pixels responsible for a smoke plume come from a single biome, and only if this condition is met do we include the fire and plume within the matchup dataset for that biome (for its $EC_{CO}^{b}$ derivation). If not we discard it from the matchup. A larger plume dataset would enable us to apply an even more stringent condition (e.g. 70% of active fire pixels from a single biome) whilst still maintaining a reasonable matchup sample size to generate $EC_{CO}^{b}$ values from for each biome, and ultimately more matchups that are even more dominated by a single biome may make the $EC_{CO}^{b}$ values more statistically distinct. In our prior study that used AOD data rather than CO data (Nguyen & Wooster 2020), the 1 km spatial resolution of the AOD product used allowed us to include more smaller plumes and smaller fires to generate the $EC_{TPM}^{b}$ values. And the fact that these plumes were generated by fires covering smaller areas meant that there were more plumes able to be included in the matchup dataset which had a larger majority of the AF pixels coming from a single biome. The resulting $EC_{TPM}^{b}$ values were indeed more statistically distinct from one another. We have re-worded the text in manuscript to explain this point more fully.

- "Remote sensing does include a measurement". Sure, but the processing of this measurement and its combination with auxiliary information and models for remote sensing is not a measurement. I can understand that the word is kept occasionally for simplicity, but it should not be used in the title.

We have amended the manuscript title and scaled down its use throughout the manuscript

- There are a few "Sentinal" left in the text.

These have now been amended